# Heavy fermions vs doped Mott physics in heterogeneous Ta-dichalcogenide bilayers

Lorenzo Crippa [1,8] ✉, Hyeonhu Bae [2,8], Paul Wunderlich[3,8], Igor I. Mazin [4,5], Binghai Yan [2], Giorgio Sangiovanni [1], Tim Wehling[6,7] & Roser Valentí [3] ✉

Controlling and understanding electron correlations in quantum matter is one of the most challenging tasks in materials engineering. In the past years a plethora of new puzzling correlated states have been found by carefully stacking and twisting two-dimensional van der Waals materials of different kind. Unique to these stacked structures is the emergence of correlated phases not foreseeable from the single layers alone. In Ta-dichalcogenide hetero-structures made of a good metallic "1H"- and a Mott insulating "1T"-layer, recent reports have evidenced a cross-breed itinerant and localized nature of the electronic excitations, similar to what is typically found in heavy fermion systems. Here, we put forward a new interpretation based on first-principles calculations which indicates a sizeable charge transfer of electrons (0.4-0.6 e) from 1T to 1H layers at an elevated interlayer distance. We accurately quantify the strength of the interlayer hybridization which allows us to unambiguously determine that the system is much closer to a doped Mott insulator than to a heavy fermion scenario. Ta-based heterolayers provide therefore a new ground for quantum-materials engineering in the regime of heavily doped Mott insulators hybridized with metallic states at a van der Waals distance.

$TaCh_2$ (Ch = S, Se) layered materials are rather distinctive in the family of transition metal dichalcogenides. While in the trigonal prismatic setting (hexagonal, H) $TaCh_2$ is a metal similar to its exceptionally well studied sister $NbCh_2$, in the octahedral setting (tetragonal, T) (Fig. 1a) $TaCh_2$ develops a highly unusual $\sqrt{13} \times \sqrt{13}$ superstructure, which effectively partitions the structure into so-called "Star of David" (SoD) clusters, each holding just one electron (1 e), mostly localized on the central Ta ion[1-4], as shown in Fig. 1b. The intralayer hopping amplitude between these localized electrons in $1T$-$TaCh_2$ is extremely small, of the order of 10 meV. Actually, in the bulk material the intercluster hopping mainly proceeds via interlayer hopping, generating a band width of the order of 100 meV. Even this band width is smaller than the effective Hubbard interaction on a cluster ($U \sim 100–300$ meV), which

induces a Mott transition in the half-filled valence band[5-7]. In the monolayer $1T$-$TaCh_2$ this path does not exist, so that the band width is anomalously small and localization strong[8] even compared to materials containing 4f electrons. This suggests nontrivial, and qualitatively stronger correlation effects in a single layer as compared to the bulk, and tempts an analogy with Kondo physics−usually manifested in 4f systems−if the monolayer is put in contact with delocalized carriers.

The above consideration was recently brought into limelight by several experimental studies on hetero-bilayers of $1T$-$TaCh_2$, a strongly Mott insulating system, and $1H$-$TaCh_2$, a good two-dimensional metal, where strong changes were observed, compared to an isolated $1T$-$TaCh_2$ layer, with a well-defined Mott gap being supplanted or augmented by a narrow zero-bias peak[9-12]. A natural interpretation,

[1]Institut für Theoretische Physik und Astrophysik and Würzburg-Dresden Cluster of Excellence ct.qmat, Universität Würzburg, 97074 Würzburg, Germany. [2]Department of Condensed Matter Physics, Weizmann Institute of Science, 7610001 Rehovot, Israel. [3]Institut für Theoretische Physik, Goethe Universität Frankfurt, am Main, Germany. [4]Department of Physics and Astronomy, George Mason University, Fairfax, VA 22030, USA. [5]Quantum Science and Engineering Center, George Mason University, Fairfax, VA 22030, USA. [6]I. Institute of Theoretical Physics, University of Hamburg, Notkestrasse 9, 22607 Hamburg, Germany. [7]The Hamburg Centre for Ultrafast Imaging, Luruper Chaussee 149, D-22761 Hamburg, Germany. [8]These authors contributed equally: Lorenzo Crippa, Hyeonhu Bae, Paul Wunderlich. ✉e-mail: lorenzo.crippa@physik.uni-wuerzburg.de; valenti@itp.uni-frankfurt.de

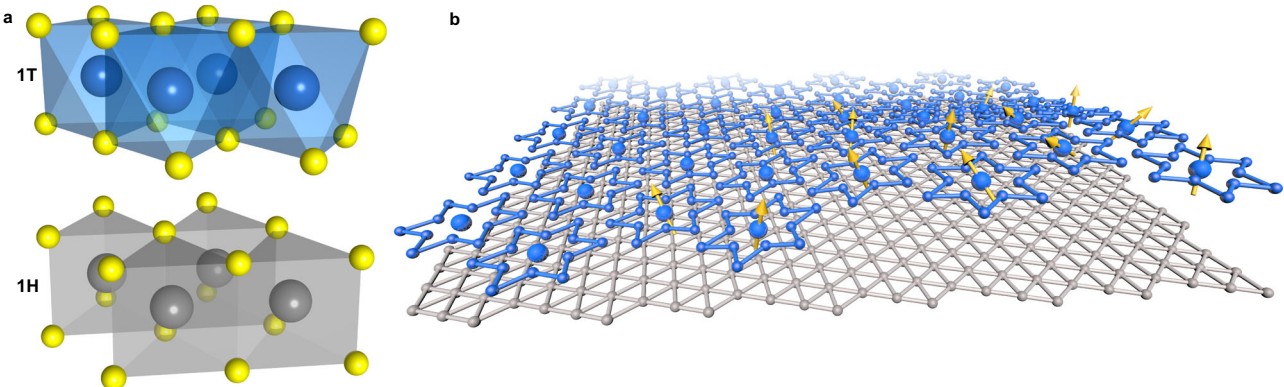

**Fig. 1 | Illustrations of the 1T/1H-TaS$_2$ bilayer structure. a** Local atomic structures for the 1T and 1H phases. Large spheres in blue and gray represent Ta atoms in 1T and 1H, respectively. Small yellow spheres denote S atoms. **b** 1T-TaS$_2$ develops a Star-of-David charge density wave pattern (blue) forming a triangular lattice. One electron is localized at the center to the Star-of-David in a free standing 1T layer. The 1H layer (gray) is metallic but hosts electrons transferred from the 1T layer via interlayer interaction. Only Ta atoms are shown here while S atoms are omitted for clarity.

invoked in these works, is in terms of conduction electrons in 1H-TaCh$_2$ screening the localized electrons in 1T-TaCh$_2$. However, this interpretation basically assumes that the screening capacity of these spatially detached carriers is comparable to that of classical Kondo systems, where the localized and itinerant electrons occupy the same space[13–16]. On the other hand, a recent first-principles study of the similar three-dimensional heterostructure, known as 4Hb-TaS$_2$, consisting of alternating layers of 1T- and 1H-TaS$_2$[17], found a strong charge transfer from the 1T to the 1H layer, rendering the valence states in the former completely depleted. Needless to say, this picture with an empty 1T-band does not leave room to any strong-correlation effects.

Here, we present a way out of this conundrum by combining ab initio density functional theory (DFT) calculations for 1T/1H-TaS$_2$ bilayers with many-body dynamical mean field theory (DMFT) calculations of the resulting low-energy models. We investigate the electronic structure of 1T/1H-TaS$_2$ bilayers (Fig. 1a, b) at various interlayer distances, $d_{int}$, since different experiments are liable to have different 1T/1H-TaS$_2$ bilayer spacing $d_{int}$. Indeed, mechanical stacking of single layers is extremely sensitive to the manufacturing details and $d_{int}$ is nearly always larger than the optimum spacing for an ideal epitaxy. In fact, ref. 11 mentions STM steps of ≈6.2 Å, which can be taken as an estimate of $d_{int}$, while the color height map shown in ref. 10 suggests $d_{int}$ ~ 8.5 Å. This is to be compared to cleaved 4Hb-TaS$_2$ bulk samples where $d_{int}$ ≤ 5.9 Å[17–19].

## Results and discussion

Our DFT results show that, while some charge transfer between 1T- and 1H-TaS$_2$ layers is very robust and unavoidable, the actual amount is highly sensitive to the separation between the layers and local environments. These findings have important consequences; an ideal bilayer system would share the same interlayer separation in the overall region as the bulk crystal 4Hb-TaS$_2$ and therefore have complete charge transfer between layers. However, in actual fabrications, as listed above, the interlayer separation is likely to be larger than in bulk 4Hb-TaS$_2$. Larger separation may accompany a decrease in charge transfer and a decrease in hybridization compared with the optimum system. Whether the electronic bands of the 1T-electrons are completely or partially empty and how correlation effects develop will be dictated by the interlayer distance between 1T- and 1H-TaS$_2$ layers.

In principle, another contributing factor to the size of charge transfer could be the presence of a substrate on top of which the bilayer is grown[10–12]. As detailed in Supplementary Note 2, a graphene substrate such as that of ref. 10 has a doping effect on the bilayer, which is however weak and rather independent on the substrate size.

To resolve the origin of the measured tunneling spectra[9–12] we investigate then in the framework of DMFT the electronic properties of the 1T/1H-TaS$_2$ bilayers by considering the low-energy models extracted from the DFT calculations at interlayer distances $d_{int}$ = 6.3 Å, 6.5 Å and 7.0 Å, which, we believe, realistically reflect the range of bilayer spacings in refs. 10,11. In this entire range, DFT yields an occupation of the 1T band of 0.4 −0.6 e. The main conclusions are: (1) screening by the 1H-TaS$_2$ layer, contrary to suggestions in previous works, is a rather minor effect; (2) the charge transfer between 1T-TaS$_2$ and 1H-TaS$_2$ drives the Mott insulating state in 1T-TaS$_2$ far away from the half-filling regime (1 e per correlated orbital) of the 1T single layer, introducing itinerant charge carriers *inside* the 1T-TaS$_2$ layer. These provide the metallic screening and lead to a zero-bias peak. The resulting state can be viewed as a strongly doped Mott insulator. Hence, the role of 1H-TaS$_2$ is not primarily screening, as originally assumed, but providing a charge reservoir, taking away some electrons from the 1T-TaS$_2$.

According to our interpretation, these TaCh$_2$ heterostructures can hence be viewed as a new platform from where to explore heavily doped Mott physics. This is particularly attractive for strongly correlated materials, as these are more standardly synthesized with integer or close-to-integer filling due to growth's complications, instabilities towards phase separation and general hostility towards large concentrations of chemical dopants. Exceptions to this scenario have often been accompanied by exotic collective phenomena, as in the cases of high-$T_c$ cuprates and iron-bansed superconductors. The case of Ta-bilayers is special for the intrinsic robustness in which the doping mechanism is realized and for the ramifications that the interplay between charge transfer and interlayer hybridization can have for materials engineering.

The optimized interlayer distance between 1T and 1H layers is $d_{int}$ = 5.81 Å (see Methods section for details), which is close to the experimental value (~5.90 Å) of the bulk 4Hb-TaS$_2$[18,20]. The band structure of this bilayer exhibits two empty, degenerate flat bands at 0.1 eV above the Fermi energy, as shown in Fig. 2a, which originate from a half-filled flat band of monolayer 1T-TaS$_2$. This is consistent with previously measured $dI/dV$ spectra of the 1T-TaS$_2$ layer in the cleaved 4Hb-TaS$_2$ sample[17–19] where 1e is transferred from the 1T to the 1H layer. After increasing the interlayer distance to $d_{int}$ = 6.3 Å, the charge transfer is reduced to 0.6 e, and the spin-polarized band structure shows a partially filled spin-majority lower band, with the spin-minority upper band being empty, as shown in Fig. 2b. Additionally, we stress the significance of the van der Waals interaction in calculations on the bilayer. For example, one can obtain an artificially enlarged optimum interlayer distance $d_{int}$ = 6.8 Å without including a van der Waals correction.

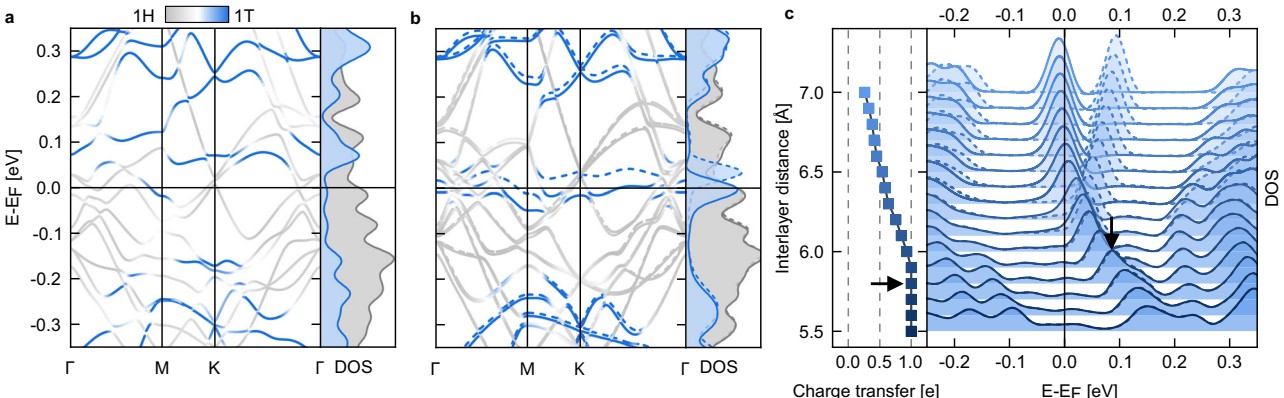

**Fig. 2 | Interlayer distance-dependent band structures, density of states (DOS), and charge transfer in the 1T/1H-TaS$_2$ bilayer. a, b** Orbital-projected electronic band structure at the interlayer distance $d_{int}$ = 5.8 Å (optimum) and $d_{int}$ = 6.3 Å (stretched), respectively. Gray and blue colors indicate the contribution from 1H and 1T layers, respectively. The blue peak in the DOS at 0.1 eV in (**a**) corresponds to empty flat bands of the 1T layer. This peak splits with the lower peak being partially filled when the interlayer distance increases to 6.3 Å as shown in (**b**). The dashed and solid lines represent opposite spin channels in DOS and band structures. **c** Amount of electrons transferred from 1T to 1H (per $\sqrt{13} \times \sqrt{13}$ supercell) for a range of $d_{int}$. The DOS of the 1T-TaS$_2$ layer for each corresponding $d_{int}$ is shown in arbitrary units; the spectra are vertically offset for clarity. Majority and minority spins are drawn in solid and dashed lines, respectively. The system at the ideal optimum interlayer distance $d_{int}$ = 5.8 Å, which was obtained by considering van der Waals corrections, is indicated by black arrows.

Figure 2c indicates the systematic distance-sensitive evolution of the charge transfer and the 1T flat band occupation. By increasing the interlayer distance, the density of states (DOS) of the 1T layer shows a continuous change. The flat band peak moves toward the Fermi energy and splits as soon as the spin-majority band starts to host a portion of the electron at distances larger than $d_{int}$ = 6.0 Å. The peak splitting increases as electron occupation grows. For the layer separation from $d_{int}$ = 5.5–7.0 Å, the charge transfer, CT, decreases from 1 to 0.4 e and the flat band filling factor increases accordingly from 0 to 0.6 e. The hybridization between the 1T-electrons and the 1H electrons decreases as well with the increase of $d_{int}$. Increasing further $d_{int}$ will lead to no charge transfer with a flat band filling factor of 1 (CT = 0) corresponding to uncoupled monolayers.

This behavior can be understood from the fact that at moderate distances there are two factors contributing to charge transfer: First, there is a chemical potential (work function) mismatch that favors some charge flow from the 1T to 1H layer; this factor is rather weakly $d_{int}$-dependent, and at $d_{int} \gtrsim 6.5$ Å it is essentially the only factor. However, at small distances, there is substantial hybridization between the flat band of the 1T layer and the conduction band of the 1H layer (whose center of gravity, as seen from Fig. 2a, b, is below the flat band), which pushes the flat band up (Fig. 2c). This effect is, on the contrary, strongly $d_{int}$-dependent and kicks in for $d_{int} \lesssim 6.5$ Å. Finally, for $d_{int} \gg 6.5$ Å (several times greater), the simple planar-plate capacitor effect, which adds additional energy cost of $\sigma d_{int}$, where $\sigma$ is the surface charge density induced by the charge transfer, arrests and prevents any further charge transfer despite the difference in the chemical potentials, and the bilayer, for all intents and purposes, become two uncoupled single layers. More details on the interdependence of charge transfer and $d_{int}$ can be found in Supplementary Note 3.

Clearly, Mott-Hubbard or Kondo type correlation effects are nonexistent for an empty 1T band, i.e., zero filling. Thus, correlation effects are only expected to play any role at all for interlayer separations exceeding $d_{int}$ = 6.0 Å when the 1T band has non-zero filling. In the following, we perform DMFT calculations to assess which kind of electron correlation effects can emerge at sufficiently large interlayer separations in the 1T/1H-TaS$_2$ bilayer. To this end, we derive a single-particle Hamiltonian from our DFT calculations (see Methods section), which describes electrons localized at each SoD ($\sqrt{13} \times \sqrt{13}$ supercell) forming a triangular lattice in the 1T layer hybridizing with a wide $d$-derived band from the 1H layer (Fig. 1b). Considering only the former as interacting degrees of freedom, we define a Periodic Anderson

Model[21,22] consisting of (1) a weakly dispersive orbital with a band width of the order of 4 meV and a local Coulomb interaction $U$ of about 100 meV (1T layer), (2) a conduction band with roughly 40-times larger dispersion (1H layer) and, (3) a hybridization $V_1$, which we assume to be local, between the localized orbital in the 1T layer and the conduction band in the 1H layer, that varies depending on the interlayer distance. A discussion on the effects of non-local interaction terms, not included in this model, can be found Supplementary Note 5. The value obtained for an interlayer distance $d_{int}$ = 6.5 Å is $V_1$ = 60 meV (see Methods). This, together with the corresponding charge transfer (CT) of 0.5 (see Fig. 2c), i.e., an average filling of the correlated orbital of 0.5 e, will be our reference parameters for the 1T/1H-TaS$_2$ bilayer.

Electron-electron interaction promotes local moments on the 1T layer and here we are interested in quantifying and determining the origin of their screening. DMFT can answer this question as it maps the problem onto a self-consistently determined impurity model in which local many-body effects of the 1T-electrons are described via a frequency-dependent (real and imaginary) self-energy. Keeping the 1H-bands explicitly in the low-energy model allows us to disentangle the two independent sources of screening for the 1T-local moments: (i) the direct 1T-1H hybridization ($V_1$), and (ii) hopping to neighboring SoD on the 1T plane. In particular, being able to self-consistently describe with DMFT the relative charge balance between 1T and 1H puts us in the position of evaluating whether the screening of the local moments in the 1T/1H-TaCh$_2$ heterostructures comes primarily from the interlayer hybridization−mechanism (i)−or from doping the SoD Mott insulator −mechanism (ii).

In Fig. 3, we show the local static spin susceptibility $\chi^{loc}_{spin}(\omega = 0) = \int_0^\beta d\tau \chi^{loc}_{zz}(\tau)$, where $\chi^{loc}_{zz}(\tau) = g^2 \langle S_z(\tau) S_z(0) \rangle$ is the static component of the spin-spin response function, with $S_z$ being the $z$-component of the spin-operator at the 1T-correlated site. The case of charge transfers CT = 1/2 and 0 are compared in Fig. 3 for various local hybridization values $V_1$, as a function of temperature. When the screening is poor, the local static spin susceptibility is expected to display a Curie-like behavior ($-1/T$). However, whenever one of the two mechanisms above starts to have appreciable effects, $\chi^{loc}_{spin}(\omega = 0)$ will deviate from $-1/T$ and gradually crossover to a flat Pauli-like response. This is more distinctly seen by fitting the data to the Curie-Weiss expression $\mu^2_{eff}/3(T + 2T_\odot)$ (solid lines in the main panels of Fig. 3), where $\mu_{eff}$ and $T_\odot$ are estimates for the static effective moments and for the screening temperature scale, respectively[23–27]. At CT = 0 (half-filling), $T_\odot$ can be identified with the Kondo temperature $T_K$[23]. This form is

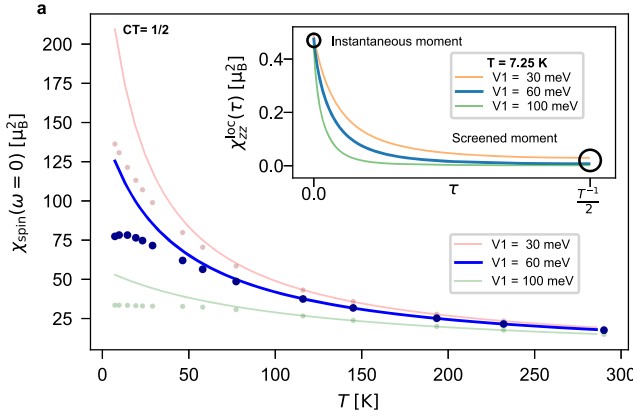

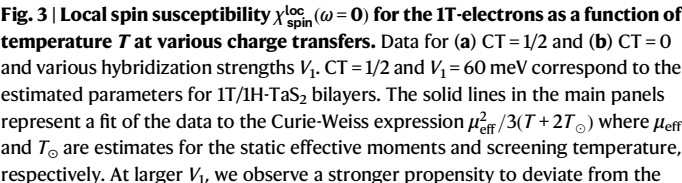

**Fig. 3 | Local spin susceptibility $\chi_{spin}^{loc}(\omega = 0)$ for the 1T-electrons as a function of temperature $T$ at various charge transfers.** Data for (**a**) CT = 1/2 and (**b**) CT = 0 and various hybridization strengths $V_1$. CT = 1/2 and $V_1$ = 60 meV correspond to the estimated parameters for 1T/1H-TaS$_2$ bilayers. The solid lines in the main panels represent a fit of the data to the Curie-Weiss expression $\mu_{eff}^2/3(T + 2T_\odot)$ where $\mu_{eff}$ and $T_\odot$ are estimates for the static effective moments and screening temperature, respectively. At larger $V_1$, we observe a stronger propensity to deviate from the

-1/$T$-behavior, characteristic of unscreened local moments. In the insets we show the imaginary time $\tau$ dependence of the local spin susceptibility at 7.25 K. While for (**a**) CT = 1/2 already at $V_1$ = 60 meV the long-$\tau$ moment is fully screened below a Curie-to-Pauli crossover temperature of about 50 K, for (**b**) CT = 0 the screening of the instantaneous ($\tau$ = 0) moment is sizeable only for the unrealistically large value of the hybridization $V_1$ = 100 meV.

however useful also away from half-filling, where despite charge fluctuations spoiling the conventional Kondo picture, a well-formed local moment $\mu_{eff}$ and the screening thereof below $T_\odot$ are clearly suggested by our data.

The fitted values of $\mu_{eff}$ and $T_\odot$ for various hybridizations are summarized in Table 1. At CT = 1/2 (Fig. 3 a) we obtain $\mu_{eff} \sim 1.23\mu_B$ for all three values of $V_1$ considered, what implies that quantum fluctuations provide a reduction of roughly 30% of the local moment of an ideally isolated spin-1/2 atom ($\sqrt{3}\mu_B$). At small $V_1$, our estimate of $T_\odot$ is of the order of 10 K and hence falls in a relevant temperature range for the physics of 1T/1H TaCh$_2$ bilayers[9,10]. Its value gets proportionally larger if we consider larger $V_1$ and continues to grow even at values of $V_1$ surpassing those suggested by our DFT analysis. This is expected, as the screening becomes more effective upon increasing the hybridization and one needs to reach higher temperatures to uncover the Curie-behavior of $\chi_{spin}$.

The effectiveness of the screening of the local moment at CT = 1/2 is evident also from the behavior of the local susceptibility with (imaginary) time, shown in the inset to Fig. 3a for a fixed temperature $T$ = 7.25 K. Because of the doping of the 1T-orbital, charge fluctuations are sizeable. Their effect is that, even for the smallest values of $V_1$, the long-$\tau$ moment (i.e., for $\tau = 1/(2T)$) is efficiently screened at this low temperature. If we compare this to the inset to Fig. 3b, i.e., to the hypothetical case of CT = 0 (no charge transfer, half-filled 1T-band, large interlayer distance limit), we can immediately assess the relative importance of the 1T/1H hybridization $V_1$ versus the 1T doping. At CT = 0, the 1T/1H hybridization is absolutely crucial to obtain a visible screening of the local moment. A sizeable reduction at large $\tau$ with respect to the instantaneous value is now visible only at exaggeratedly large $V_1$. For the realistically extracted $V_1$, the estimates of $T_\odot$ are below

1 K and $\mu_{eff}$ is only marginally reduced from the atomic value. The corresponding strong Curie-like behavior of the static susceptibilities can be seen in the main panel of Fig. 3b. Further insight on the difference between doped-Mott and heavy-fermions scenarios can be gained by analyzing the Fermi surfaces and momentum-resolved spectral functions, as done extensively Supplementary Note 6. In the first case, 1T coherent features, due to intra-layer hopping, are present, though with different weight, across the whole temperature range; in the second, coherence, this time due to interlayer hybridization, is developed only at low temperature (-10 K), while at high-T (-100 K) the 1T spectral weight is incoherently distributed.

These results displayed in Fig. 3a, b, together with the ones found in the Supplementary Note 6, demonstrate that the (hole)-doping of the 1T layer is therefore the main driver of the screening processes in 1T/1H TaCh$_2$ bilayers.

In Fig. 4, we show the temperature evolution of the spectral function at CT=1/2 for the three different values of the hybridization that we have calculated. We find that at $V_1$ = 60 meV (Fig. 4b), the low-frequency spectral features can be traced back to the coherent peak visible already at CT = 0 (see inset of Fig. 4). This value of the 1T/1H hybridization represents, therefore, an intermediate situation between the case with almost absent metallic peak of $V_1$ = 30 meV (Fig. 4a) and the unrealistically large value of $V_1$ = 100 meV (Fig. 4c) supporting a highly effective screening.

The evolution of a narrow coherence peak observed in spectra cannot alone distinguish between the doped Mott and the heavy fermion scenario: indeed, the presence of a 1H-dip in spectral weight, insensitive to transverse magnetic fields has been put forward[10] as an argument against the superconductive or CDW origin of the gap and in favor of a heavy fermion interpretation. An equally plausible explanation of the combined 1T-peak and 1H-depletion could be based on a generic Fano resonance[28], be it of Kondo origin or otherwise. The origin of the 1H-dip warrants further investigation, and is outside the scope of this paper. However, all hybridizations considered here have unavoidably charge transfer, as our DFT simulations demonstrate. The charge transfer boosts quasi-particle formation at low-energy at all hybridizations. This is why the combined DMFT and DFT results presented here, support a "doped-Mott" scenario for understanding the scanning tunneling microscopy/spectroscopy experiments[9-12] in TaCh$_2$ bilayers in these ranges of temperature.

We note that experimental spectra reported for 1T/1H TaCh$_2$ bilayers consistently feature temperature dependent "coherence"

**Table 1 | Fitting parameters of the Curie-Weiss expression for susceptibilities at different values of charge transfer (CT) and interlayer distance, on which the potential $V_1$ depends, as detailed in the Methods section**

| $V_1$ [meV] | CT = 1/2 | | CT = 0 | |
|---|---|---|---|---|
| | $\mu_{eff}$ [$\mu_B$] | $T_\odot$ [K] | $\mu_{eff}$ [$\mu_B$] | $T_\odot$ [K] |
| 30 | 1.23 | 10.53 | 1.68 | 0.01 |
| 60 | 1.23 | 19.62 | 1.61 | 0.67 |
| 100 | 1.23 | 52.21 | 1.45 | 4.33 |

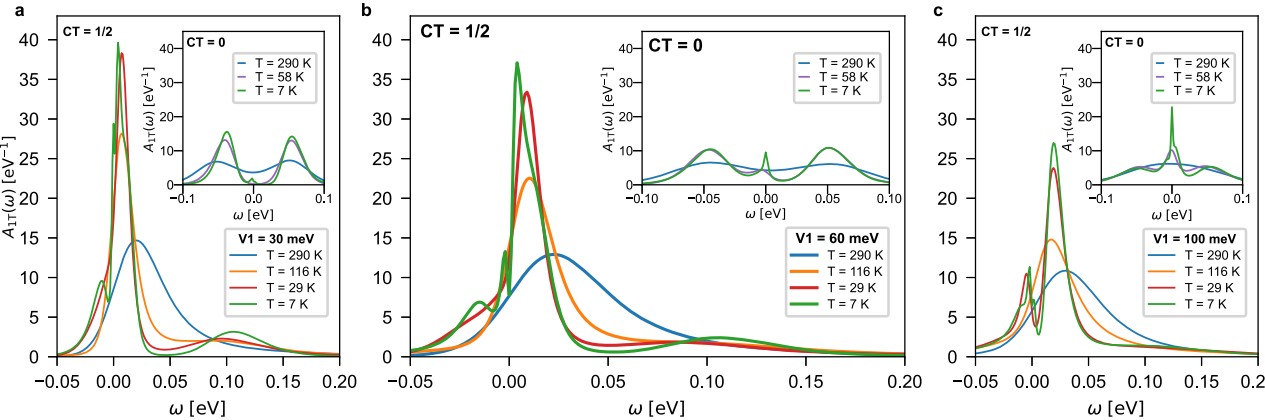

**Fig. 4 | Local spectral function of the 1T-electrons for various values of hybridization strengths.** Data for (**a**) $V_1 = 30$ meV, (**b**) 60 meV, and (**c**) 100 meV and different temperatures. The data in the main panel are for CT = 1/2 (0.5 $e$), with the corresponding CT = 0 (1e) spectral functions in the insets. Upon increasing $V_1$, at CT = 1/2 the spectrum evolves from a "peak + shoulder" structure to a pronounced "bonding/antibonding" feature. Correspondingly, at CT = 0 a coherent peak develops close to the Fermi energy for higher $V_1$. While the T/H hybridization is at the origin of such resonances in the spectrum for high values of $V_1$, that is not the case for lower values, where the zero-frequency peak can be attributed to a doped Mott insulator scenario.

peaks at the Fermi level[9–12], which is in line with our analysis. However, details like asymmetries and emergence of (pseudo)gaps vary between different experiments. This is not surprising since the interlayer distance, where the two layers are mechanically placed upon each other, is extremely sensitive to the quality of the interface and practically never can be as small an in an ideal epitaxial stacking. Actually, some of the theoretical spectra reported in Fig. 4 reveal temperature dependent fine structures as well. The fact that these fine structures are rather parameter dependent might explain the variation of spectra between different experiments[9–12].

Concluding, our results show that the 1T/1H TaCh₂ bilayer allows access to a rather little studied regime of a Mott insulator on the triangular lattice with low occupancy of less than 1/2 electron per site. In its simplest form, a doped Mott state on a triangular lattice is a prime candidate for chiral superconductivity and correlated topological states of matter. Furthermore, we have charge density wave physics and Ising superconductivity in the H-phase layer which is (albeit weakly) hybridization-coupled to the doped Mott state uncovered for the 1T/1H hybrid system. Thus, the paradigm of doped Mott physics is enriched here by the proximity coupling to further correlated quantum states, and novel quantum states by means of controlling the low-energy physics might emerge.

## Methods
### Density functional theory
We carried out DFT calculations as implemented in the Vienna ab initio Simulation Package (VASP) within the projector augmented wave method[29]. The generalized gradient approximation exchange-correlation energy functional as parameterized by Perdew-Burke-Ernzerhof[30] and the DFT-D3 method[31] were employed to describe van der Waals interactions for simulating realistic lattice parameters. Spin polarization was considered in all calculations. The kinetic energy cutoff was set to 600 eV, and electronic energy convergence was achieved when the total energy difference between steps reached less than $10^{-7}$ eV. Ionic relaxation was performed until the Hellmann-Feynman forces acting on each ion became smaller than 0.01 eV Å⁻¹.

The bilayer 1T/1H-TaS₂ was simulated in a $\sqrt{13} \times \sqrt{13}$ supercell accompanied by CDW structural modulations, to take into account the expected behavior of the 1T layer. While in principle a $3 \times 3$ CDW is also expected on the 1H layer, its presence was find to have a minimal effect on the work functions of the system, as detailed in Supplementary Note 1, and was hence discarded.

About 10 Å of vacuum slab out of the two-dimensional system was adopted to eliminate spurious interactions in the periodic cell scheme. In the calculation with increasing interlayer distances, all Ta atoms in each layer are placed on parallel planes to maintain artificial interlayer spacings, but S atoms are fully relaxed.

In principle, a DFT + U correction can impact band stacking and charge transfer; we note, however, that the related energy shift of around 100 meV would be roughly 1/6 of the T-layer bonding/antibonding energy gap (see Fig. 2b) and 1/8 of the work function difference between the 1T and 1H layer (see Supplementary Table 1), providing therefore no significant alteration to both band stacking and charge transfer.

### Tight-binding model
The tight-binding model (TB), $H_0 = H_T + H_H + H_V$, underlying the correlated electron analysis consists of three blocks, where $H_T$ describes the well-localized orbital centered at each SoD ($\sqrt{13} \times \sqrt{13}$ supercell) in the 1T layer, $H_H$ describes the wide band from the 1H layer and $H_V$ the hybridization of the two.

A free standing 1H monolayer has a single band near the Fermi level which carriers mostly $d_{z^2}$-character around Γ but $\{d_{x^2-y^2}, d_{xy}\}$-character towards the Brillouin zone boundaries. The $3 \times 3$ CDW emerging in the 1H layer suppresses the $\{d_{x^2-y^2}, d_{xy}\}$ spectral weight near the Fermi level. Moreover, out of-plane $d_{z^2}$-orbitals provide much stronger interlayer hybridization than $\{d_{x^2-y^2}, d_{xy}\}$ (see also Supplementary Note 7). Besides, the screening properties of the itinerant electrons in the 1H layer are very weakly sensitive to their orbital composition, as long as the Fermiology is correct. Thus, we focus on the $d_{z^2}$-orbitals of the 1H layer in $H_H$ and $H_V$. With the Fermi energy at $E_F = 0$, we find that the $d_{z^2}$ part of the 1H layer can be captured by a simple tight-binding model on the triangular lattice

$$H_H = \epsilon_H \sum_{i\alpha} c_{i\alpha}^\dagger c_{i\alpha} + t_H \sum_{<i\alpha,j\beta>} c_{i\alpha}^\dagger c_{j\beta}, \qquad (1)$$

with on-site energy $\epsilon_H = -370$ meV and nearest neighbor hopping $t_H = 150$ meV. Here, $i$ is a combined unit cell and spin index. The H and the T-phase have roughly the same lattice constant. We thus assume 13 Ta atoms per SoD in the H-layer. $\alpha \in 0...12$ enumerates the 13 Ta atoms in the H layer per $\sqrt{13} \times \sqrt{13}$ supercell. $<i\alpha, j\beta>$ denotes pairs of nearest neighbor orbitals with equal spin. $c_i^\dagger$ ($c_i$) denote corresponding creation (annihilation) operators.

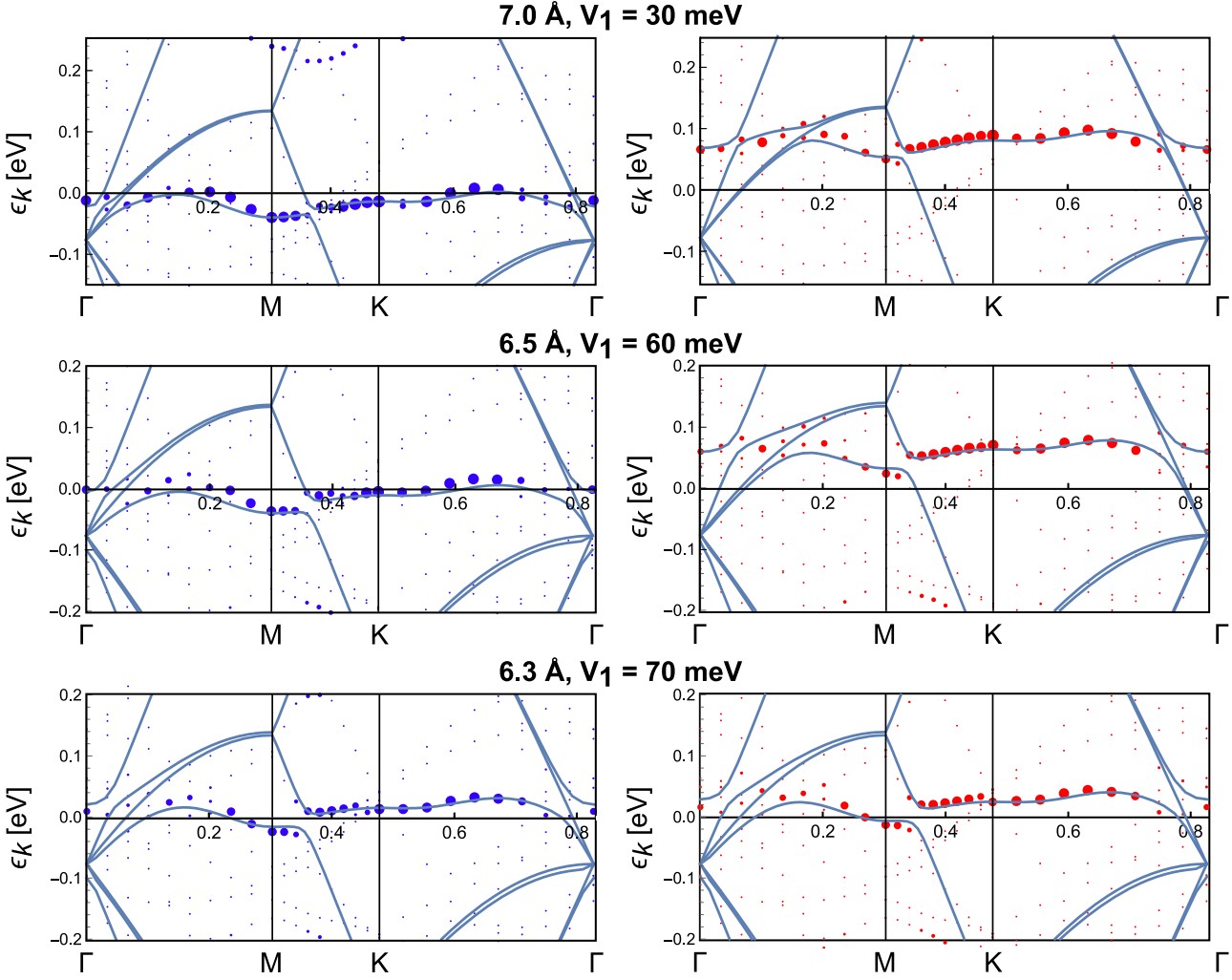

**Fig. 5 | Tight-binding fit of the DFT bands at interlayer separations.** Data for $d_{int} = 7.0$ Å (upper), $d_{int} = 6.5$ Å (middle), and $d_{int} = 6.3$ Å (lower panels). The fit is shown for the majority spin components (left column) and minority spin components (right columns). Tight-binding bands are shown as solid lines. The DFT bands are shown as dots, where the dot size visualizes the $d_{z^2}$ orbital weight from the T-layer. The fitted hybridizations between T- and H-layer, $V_1$, are indicated in the respective rows. The on-site energies for majority (minority) spin found in the TB fits are $\epsilon_T = -15$ meV ($\epsilon_T = 80$ meV) at $d_{int} = 7.0$ Å, $\epsilon_T = -15$ meV ($\epsilon_T = 60$ meV) at $d_{int} = 6.5$ Å, and $\epsilon_T = 10$ meV ($\epsilon_T = 20$ meV) at $d_{int} = 6.3$ Å.

To describe the 1T-derived flat band and its interlayer distance $d_{int}$-dependent coupling to the 1H electrons, we analyze DFT band structures at $d_{int} = 7.0$ Å, 6.5 Å, and 6.3 Å (see Fig. 5). At the largest separation, $d_{int} = 7.0$ Å, we fit the 1T layer flat band with a third nearest neighbor tight-binding model

$$H_T = \epsilon_T \sum_i d_i^\dagger d_i + t_{T1} \sum_{\langle i,j \rangle} d_i^\dagger d_j + t_{T2} \sum_{\langle\langle i,j \rangle\rangle} d_i^\dagger d_j + t_{T3} \sum_{\langle\langle\langle i,j \rangle\rangle\rangle} d_i^\dagger d_j \quad (2)$$

yielding hoppings $t_{T1} = 2.1$ meV, $t_{T2} = -0.8$ meV and $t_{T3} = -3.75$ meV (compare to $t_H = 150$ meV). While our fit also yields on-site energies given in Fig. 5, these will not enter our DMFT simulations, since we treat $\epsilon_T$ as adjustable parameter to fix the occupation of the band derived from the T-layer. Note that in the 1T layer we have one orbital per $\sqrt{13} \times \sqrt{13}$ supercell in our model.

We assume that hybridization between states in the H-layer and the flat band in the T-layer takes place via the H-layer Ta atom ($\alpha = 0$),

which is directly underneath the SoD center in the T-layer:

$$H_V = V_1 \sum_i d_i^\dagger c_{i,0} + h.c. \quad (3)$$

By fitting to our DFT calculations we obtain $V_1 \approx 30$ meV at interlayer spacing of $d_{int} = 7.0$ Å, $V_1 \approx 60$ meV at $d_{int} = 6.5$ Å, and $V_1 \approx 70$ meV at $d_{int} = 6.3$ Å.

The underlying DFT data are spin-polarized, while the TB matrix elements as fitted here are the same for majority and minority spin. In a first step, the on-site energies come out spin-dependently in the fits. However, we are disregarding this initial spin-dependence in DMFT, where we put a spin-averaged on-site energy (plus double counting correction) for the 1T layer electrons.

### Dynamical mean-field theory
We account for correlation effects between electrons in the flat 1T-derived band by supplementing the previously detailed tight-binding

model with the standard interaction term

$$H_{\text{int}} = U \sum_i \hat{n}_{i\uparrow} \hat{n}_{i\downarrow} \tag{4}$$

where $\hat{n}_{i\sigma} = d_{i\sigma}^\dagger d_{i\sigma}$ and U is the Hubbard local two-body repulsion potential, set in our case to 100 meV. Via DMFT, we can nonperturbatively describe all the local quantum fluctuations of the correlated system by mapping the low-energy subspace to an Anderson impurity model featuring a self-consistently determined bath. We ran the DMFT simulations with the CTQMC-CTHYB software suite *w2dynamics*[32]. The picture painted by the tight-binding model is that of a system consisting of a correlated orbital and 13 uncorrelated spectator ones. We ran our simulations in the paramagnetic phase. Since the tight-binding model includes both correlated and uncorrelated orbitals, a double counting correction has to be added to the single-particle Hamitonian in the form of a shift of the chemical potential for the correlated subspace. We adjust such shift self-consistently at each step of the DMFT loop, to ensure the occupation of the correlated subspace is as requested (quarter-filled for charge transfer 1/2 and half-filled for charge transfer 0). The Quantum Monte Carlo DMFT solver gives direct access to all local dynamical self-energies and response functions on the imaginary time/Matsubara frequency axis, hence the spin susceptibility can be directly determined from

$$\chi_{zz}^{\text{loc}} = g^2 \langle S_z(\tau) S_z(0) \rangle \tag{5}$$

where $S_z(\tau) = (n_\uparrow(\tau) - n_\downarrow(\tau))/2$. The local static spin susceptibility fit via the Curie-Weiss formula $\mu_{\text{eff}}^2/3(T + 2T_\odot)$ gives the results summarized in Table 1.

## Data availability
The band, DOS, susceptibility, and spectral function data visualized in the figures, as well as the input parameters and necessary files to replicate the DFT and DMFT runs, are available in the WueData repository with the identifier https://doi.org/10.58160/125[33].

## Code availability
The DFT simulations have been obtained using the VASP software suite[29]. The DMFT simulations were obtained using the w2dynamics CT-HYB solver[32].

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

## Acknowledgements

We thank N. Avraham, J. Ruhman, A. Keselman, I. Kimchi, B. Kalisky, L. de' Medici, and A. Toschi for useful discussions. We acknowledge support from the Deutsche Forschungsgemeinschaft (DFG, German Research Foundation) through QUAST FOR 5249 (Project No. 449872909, projects P4 and P5), the Cluster of Excellence 'CUI: Advanced Imaging of Matter'—EXC 2056 (Project No. 390715994), and SPP 2244 (WE 5342/5-1 project No. 422707584). B.Y. acknowledges the financial support by the European Research Council (ERC Consolidator Grant "NonlinearTopo", No. 815869) and the ISF—Personal Research Grant (No. 2932/21). I.I.M. was supported by Office of Naval Research through grant N00014-23-1-2480, and is grateful to the Wilhelm and Else Heraeus Foundation for supporting his visit to University of Frankfurt. L.C. and G.S. were supported by the Würzburg-Dresden Cluster of Excellence on Complexity and Topology in Quantum Matter -ct.qmat Project-ID 390858490-EXC 2147, and gratefully acknowledge the Gauss Centre for Supercomputing e.V. (www.gauss-centre.eu) for funding this project by providing computing time on the GCS Supercomputer SuperMUC at Leibniz Supercomputing Centre (www.lrz.de). L.C. gratefully acknowledge the scientific support and HPC resources provided by the Erlangen National High Performance Computing Center (NHR@FAU) of the Friedrich-Alexander-Universität Erlangen-Nürnberg (FAU) under the NHR project b158cb.

## Author contributions

All authors made contributions to the development of the approach and wrote the paper. HB and IIM performed the DFT calculations. LC and PW performed the DMFT calculations. TW performed the TB calculations. BY, IIM, TW, GS, and RV supervised the project.

## Funding

## Competing interests

The authors declare no competing interests.
