## [Peer Review File · Nature Communications]

REVIEWER COMMENTS

Reviewer #1 (Remarks to the Author):

The authors provide an interesting first-principles study of 1H/1T TaCh₂ heterostructures. Based on their density functional theory calculations, the authors show that there is a sizable charge transfer from 1T to 1H, suggesting that the system behaves in the doped Mott regime. Using a low-energy model fitted to the DFT band structure, the authors perform DMFT calculations showing that a peak close to zero energy appears as observed in experiments.

I find their results are of potential interest for experiments in 1T/1H multilayers and can motivate further theory work in related systems. Their manuscript is certainly of interest to the first principles community of 2D materials and strongly correlated physics. The manuscript is well written, the figures are of good quality, and the numerical results are correct within the approximations made in the manuscript. Before recommending the manuscript for acceptance in Nature Communications, I think that there are several points that the authors should address.

- In 4Hb heterostructures, there is experimental evidence that local replacement of S by Se drives the local charge density wave from the empty state (noted by the authors in the introduction) towards half filling. Would the results of the authors be consistent with this difference in charge transfer between Se and S?

- Both the 1T and 1H layers show a charge density wave. If I understand correctly, in the calculations, only the 1T charge density wave is included. This is, of course, understandable due to commensurability issues. Nonetheless, I would like to ask the authors if they quantified in some way the error induced by this assumption to have an estimate of how much this could affect charge transfer.

- The authors perform DFT calculations, focusing on the charge transfer between 1T and 1H. Charge transfer can be strongly dependent on DFT+U correction in the d-orbitals of Ta, specifically in the 1T layer. Have the authors considered the impact of a DFT+U correction in their results?

- Following up on the previous point, first principles methods often severely misestimate interactions gaps, both due to inaccurate treatment of local and long-range interactions. Not treating long-range interactions accurately is, of course, also understandable, but I think that it could be beneficial briefly commenting on it. Could the authors comment on how a more accurate treatment of long-range interactions could modify their results?

- The authors perform DMFT calculations including a local U in a low-energy model fitted to the 1T bands. The Wannier orbitals of 1T are known to have a relatively delocalized nature, namely having finite weight in several Ta atoms. This implies that the first neighbor repulsion of the low-energy

model could be sizable in comparison with U . Could the authors comment and quantify what would be the impact of first neighbor repulsion in their low-energy model?

- The authors performed calculations of free-standing 1H/1T. In the experiment of Refs. 10,11,12 a substrate is present, meaning that it is assumed that the substrate does not modify the charge transfer. However, in my understanding, substrate effects can substantially change band alignment, thus modifying charge transfer. Could the authors show some theoretical evidence that a substrate would not change charge transfer in this case?

- In Ref. 10 a gap is observed when tunneling onto the 1H layer. Is this gap also rationalized in terms of the calculations of the authors, or would the picture of doped Mott only account for the zero energy peak when the 1T layer is probed?

To summarize, I think that their results are of interest to the first principles 2D materials community and can be of relevance for experiments in the field. Their manuscript provides a potentially useful reference for future first principles theoretical treatments, and therefore I believe that it will be a valuable addition to the literature. Given all the points above, if the authors address the points mentioned above, I would recommend their manuscript for acceptance in Nature Communications.

Reviewer #2 (Remarks to the Author):

The authors investigate the properties of heterogeneous Ta-dichalcogenide bilayer addressing the character of the metallic state obtained interfacing 1T-TaCh₂ insulating and 1H-TaCh₂ metallic layer. Recent experiments claimed that Kondo screening leads to the screening of the local moments of the strongly correlated insulating layer due to the delocalized electrons giving rise to an heavy Fermi liquid. This observation is contrasted by recent studies of in 3D heterostructure consisting of alternating layers of 1T-TaS₂ and 1H-TaS₂ that shows strong charge transfer from the insulating layer to the metallic one. This effect induces a considerable number of holes in the 1T-TaS₂ layer reducing the role of correlation.

The paper provides a detailed analysis based on combined DFT+DMFT simulations to settle these contrasting results. The paper confirms the importance of the interlayer charge transfer, finding that the role of the metallic layer is the one of screening the local moments but rather to provide an electronic reservoir leading to a strongly doped Mott insulator.

The paper is well written, the periodic Anderson model proposed to model the bilayer is interesting and the results sound physically correct.

However, I do have several comments that I wish the authors to consider. Please see the detailed comments below.

(1) The strong valence fluctuations leading to a Mott insulator with less than $1/2$ electron per site suggests that the interlayer Coulomb interaction between delocalized carriers in the 1H and localized one in the 1T layer plays a crucial role. Did you consider what is the role of interlayer Coulomb interaction? Do you have any estimate of the value of the interlayer interaction?

(2) Related to point (1). The nearest-neighbor Coulomb interaction leads to the formation of interlayer exciton complexes [electron in 1H bound to hole in the 1T layer] that as the interlayer distance is reduced might lead to the charge transfer instability.

Can you comment on the role of these interlayer excitations?

(3) Concerning Fig.3 DMFT simulations are performed at $CT=0$ [filling 1 of the flat band in 1T layer] and $CT=1/2$. Generically, the number of electrons in the 1T layer is not conserved due to the interlayer hybridization. How did you fix the relative filling of the two layers?

(4) Concerning your discussion on artificial Kondo physics realized by stacking different layer of TMDs there have been recent theoretical proposals and experimental realizations of heavy Fermi liquid like behavior in TMD moire' semiconductors, see Ref. below:

Th.

[1] <https://journals.aps.org/prb/abstract/10.1103/PhysRevB.106.L041116>

[2] <https://journals.aps.org/prresearch/abstract/10.1103/PhysRevResearch.3.043173>

[3] <https://www.science.org/doi/abs/10.1126/sciadv.ade7701>

Exp.

[4] <https://www.nature.com/articles/s41586-023-05800-7>

To broaden the discussion and make connection with recent advances in the field I suggest the authors to cite the aforementioned references.

Reviewer #3 (Remarks to the Author):

The present work provides insights into the physics of 1T/1H TaCh₂ heterostructures, exploring the possibility of classifying them as either heavy fermions or doped Mott insulators. The authors highlight two crucial competing parameters, namely the charge transfer and interlayer hybridization strength, and demonstrate through DFT calculations that these parameters vary with changes in interlayer distance. Furthermore, they propose that this system should be regarded as a highly doped Mott insulator rather than a heavy fermion, offering a clearer understanding of the experimental observations on 1T/1H TaCh₂ heterostructures. However, further investigations are required to determine the optimal parameters that accurately describe the Kondo behavior of this system. While this work contributes to enhancing our understanding of correlation effects induced by vdW heterostructures, it is suggested that publication in a more specialized journal would be appropriate. Several comments are provided to elucidate the authors' work:

It would be beneficial to elucidate the underlying cause of the change in charge transfer with varying interlayer distance. As my understanding, charge transfer tends to approach from ~ 0.3 at infinite distance to 1.0 as the distance decreases, at the same time interlayer interaction strengthens. This suggests the possibility of a bonding/antibonding relationship between the two layers. Is it feasible to identify the primary characteristic of the interlayer interaction?

The physical range of charge transfer and interlayer hybridization is obtained solely through DFT calculations. However, these values can be easily influenced by different calculation options. For instance, it would be valuable to know if the authors considered full structural relaxation even at short interlayer distances, as the formation of SoD may be weakened or disrupted in highly doped cases during DFT calculations. Additionally, long-range correlation effects are known to impact band positions and charge transfer.

Based on Figure 3, it seems that the susceptibilities for (CT=1/2, V1=30) and (CT=0, V1=100) exhibit similar trends upon rough comparison. Moreover, the spectral functions in Figure 4 demonstrate similar behavior, albeit with some asymmetry. If we consider different values of CT and V1 that exhibit similar low-energy behavior, is it possible to determine the specific values of CT and V1 through experimental observations? It is crucial to clarify whether this system can be classified as a heavy fermion, highly doped Mott insulator, or something in between.

LIST OF CHANGES

1. Added supplemental information containing additional information on charge density wave in the 1H layer, effects of a substrate on the system, effect of interlayer distance on charge transfer, how atomic relaxation has been accounted for in DFT simulations and analysis of Fermi surface and momentum-resolved spectral functions for the heavy-Fermions and doped-Mott scenarios.
2. Modified Fig.2 in the main text to increase resolution
3. Added references suggested by reviewer 2 at line 79
4. Added a brief discussion on the effects of a substrate, with reference to supplemental information, at lines 120-126
5. Expanded the discussion on the relation between interlayer distance and charge transfer with reference to supplemental information, lines 200-221
6. Added a sentence mentioning the treatment of 1H CDW with reference to supplemental information, lines 415-419
7. Added a sentence discussing the necessity or lack thereof of DFT+U corrections, lines 426-432
8. Added a sentence affirming the difference in spectral function and Fermi surface shape with reference to the supplemental information, lines 328-338

REPLY TO REVIEWER 1

The authors provide an interesting first-principles study of 1H/1T TaCh₂ heterostructures. Based on their density functional theory calculations, the authors show that there is a sizable charge transfer from 1T to 1H, suggesting that the system behaves in the doped Mott regime. Using a low-energy model fitted to the DFT band structure, the authors perform DMFT calculations showing that a peak close to zero energy appears as observed in experiments.

I find their results are of potential interest for experiments in 1T/1H multilayers and can motivate further theory work in related systems. Their manuscript is certainly of interest to the first principles community of 2D materials and strongly correlated physics. The manuscript is well written, the figures are of good quality, and the numerical results are correct within the approximations made in the manuscript.

We thank the reviewer for their positive assessment.

Before recommending the manuscript for acceptance in Nature Communications, I think that there are several points that the authors should address.

- 1. In 4Hb heterostructures, there is experimental evidence that local replacement of S by Se drives the local charge density wave from the empty state (noted by the authors in the introduction) towards half filling. Would the results of the authors be consistent with this difference in charge transfer between Se and S?*

We thank the reviewer for the question. Since S and Se are electronically extremely similar, we don't expect that the electronic state in question should be affected by small local displacements due to a different ionic radius of Se. Indeed,

Figure 1: **Interlayer distance-dependent flat band filling in bilayer 1T/1H-TaS₂ with Se substitutional defect.** Density of states (DOS) of the 1T-TaS₂ layer contribution when a 2% impurity concentration of Se is considered in the simulations of bilayer 1T/1H-TaS₂. The DOS is shown in arbitrary units for various interlayer distances d_{int} . The spectra are vertically offset for clarity. Majority and minority spins are drawn in solid and dashed lines, respectively. At $d_{int} \gtrsim 6.1$, a splitting of flat bands is expected as the hybridization decreases, as it is happening in 1T/1H-TaS₂ without Se substitution.

to check this, we performed calculations for the bilayer 1T/1H-TaS₂ containing a small amount of Se defects, and found rather minor changes. For the simulation, the impurity concentration was tuned to 2%, which corresponds to Ta₂₆S₅₁Se₁. Fig. 1 shows the interlayer distance dependent flat band filling for this bilayer. At the equilibrium distance $d_{int} \sim 5.8$ Å, we found degenerate states without spin splitting,

which indicate no fractional filling on the localized state. The splitting of flat bands increases at $d_{int} \gtrsim 6.1$ very similarly to the results for pure bilayer 1T/1H TaS₂ without Se impurities. These results indicate that a small Se substitution does not affect the charge transfer and hybridization in the 1T/1H-TaS₂ system. Instead, it implies that the experimentally observed partially filled state must have a different origin, such as, for instance, the presence of larger interlayer separation in the sample.

2. *Both the 1T and 1H layers show a charge density wave. If I understand correctly, in the calculations, only the 1T charge density wave is included. This is, of course, understandable due to commensurability issues. Nonetheless, I would like to ask the authors if they quantified in some way the error induced by this assumption to have an estimate of how much this could affect charge transfer.*

In our calculations, we considered a $\sqrt{13} \times \sqrt{13}$ supercell because the formation of SoD on the 1T layer was essential to observe flat band behavior. To quantify the effect of the 1H layer 3×3 CDW on the bilayer 1T/1H-TaS₂ system, we have investigated work functions of monolayer 1T- and 1H-TaS₂ with and without CDW modulation, and calculated the charge transfer in the bilayer system. We selected typical 3×3 CDW phases on 1H [1–3] as illustrated in Fig. 2. As shown in Table I, a 3×3 CDW produces no notable variations in the work function. In contrast, the formation of SoD on 1T results in 53 meV (the difference between $\sqrt{13} \times \sqrt{13}$ and 1×1 1T structures). We also performed a Bader charge analysis to investigate the effect of CDW on the charge transfer between the two layers and find it to be comparable despite the formation of distinct CDW phases. This result indicates that the 3×3 CDW on the 1H layer can be neglected in the investigation of the electronic properties of the bilayer system and only the CDW in the 1T layer plays

Figure 2: **CDW phases on 3×3 monolayer 1H-TaS₂**. Top view of atomic structures of monolayer 1H-TaS₂. Ta and S atoms are depicted as blue and yellow spheres, respectively. (a) triangular lattice in a 1×1 unit cell without CDW modulation. (b-d) selected CDW phases (#1 to #3 in Table I, respectively) on 3×3 monolayer 1H-TaS₂. Atomic displacements and unit cells are illustrated as black arrows and black dashed lines, respectively. Configuration (d) has the lowest formation energy of the selected configurations.

an essential role. In the supplementary information of the revised manuscript we summarize these results, and we reference them in the main text.

3. *The authors perform DFT calculations, focusing on the charge transfer between*

	unit cell	work function [eV]	$\Delta E/\text{f.u.}$ (meV)
monolayer 1H-TaS ₂	1 × 1	5.905	3.28
	3 × 3 CDW #1	5.902	1.43
	3 × 3 CDW #2	5.907	0.66
	3 × 3 CDW #3	5.896	0
monolayer 1T-TaS ₂	1 × 1	5.070	22.51
	$\sqrt{13} \times \sqrt{13}$ CDW	5.123	0
	unit cell	charge transfer [$e/(\text{TaS}_2)_{26}$]	$\Delta E/\text{f.u.}$ (meV)
bilayer 1T/1H-TaS ₂	1 × 1	0.44	12.3
	3 × 3 (CDW on 1H)	0.43	11.4
	$\sqrt{13} \times \sqrt{13}$ (CDW on 1T)	0.41	0

Table I: **Effect of CDW modulation on the charge transfer in bilayer 1T/1H-TaS₂** Calculated work functions of monolayer 1H-TaS₂ and 1T-TaS₂ with and without CDW modulation. As shown in Fig. 2, typical CDW phases on 3 × 3 monolayer 1H-TaS₂ were chosen [1–3]. Formation energy differences per formula unit between identical polytypes are listed on the table. $\Delta E = 0$ indicates the most stable configuration. The work function difference between the metallic 1 × 1 1T and the Mott-insulating $\sqrt{13} \times \sqrt{13}$ 1T structures is 53 meV, whereas the CDW on metallic H-TaS₂ does not produce significant differences. Also shown is the calculated charge transfer that occurs when 1T and 1H layers combine to form a bilayer structure ((TaS₂)₂₆) with CDW modulation in each layer. In spite of differences in periodicity and structural deformation, the charge transfer between the two polytypes is comparable.

Figure 3: Schematic view of the hierarchy of energies in the 1T layer.

1T and 1H. Charge transfer can be strongly dependent on DFT+U correction in the d-orbitals of Ta, specifically in the 1T layer. Have the authors considered the impact of a DFT+U correction in their results?

To answer this question, it is important to understand the hierarchy of energies in the 1T layer, as sketched in Fig.3. There are three energy scales: $t_{SoD} \ll U, E_g$, where t_{SoD} is the hopping between two neighboring Stars of David, and E_g is the energy separation between the valence and the conduction band *excluding the strongly correlated band near the Fermi level*. Because of this hierarchy, the charge transfer (CT) depends on U rather weakly (as long as it remains $\ll E_g$). Since one of our main points is that CT is very sensitive to the interlayer distance and the quality of the interface, and varies considerably from sample to sample, a small correction due to U is less important. In the revised version, we include a new sentence clarifying this aspect.

4. *Following up on the previous point, first principles methods often severely misestimate interactions gaps, both due to inaccurate treatment of local and long-*

range interactions. Not treating long-range interactions accurately is, of course, also understandable, but I think that it could be beneficial briefly commenting on it. Could the authors comment on how a more accurate treatment of long-range interactions could modify their results?

The reviewer is right that DFT (PBE or LDA) underestimates the value of the band gaps, in our case E_g between the uncorrelated bands. However, corrections to E_g are here not essential since we are interested in the interaction between the 1H layer (good metal, no corrections) and the Mott 1T band (where the role of the U correction has been discussed above). To see these statements more quantitatively, in Fig. 4 we investigated the DOS of the 1T layer at various levels of exchange correlation functionals. Within PBE+U, r²SCAN+rVV10 [4], and local-mBJ [5]. Only the value $U_{eff} = 2.27$, calculated using the linear response method for Ta-5d orbitals [6], was used in the PBE+U calculation. r²SCAN+rVV10 and local-mBJ functionals were selected due to their strength in the implementation of vdW interaction and non-local exchange. Although there are slight variations in the magnitude of the peak splitting, the interlayer spacing where the splitting begins, and the shape of flat band peaks, we observe that the overall trends are not significantly changed in comparison to pure PBE results.

- 5. The authors perform DMFT calculations including a local U in a low-energy model fitted to the 1T bands. The Wannier orbitals of 1T are known to have a relatively delocalized nature, namely having finite weight in several Ta atoms. This implies that the first neighbor repulsion of the low-energy model could be sizable in comparison with U. Could the authors comment and quantify what would be the impact of first neighbor repulsion in their low-energy model?*

We thank the reviewer for this important question. While we cannot calculate the intercluster Coulomb repulsion from first principles, we can do a back-of-the

Figure 4: **Interlayer spacing-dependent evolution of the Projected Density of States of the 1T-TaS₂ layer at the GGA and meta-GGA level of theories.** DOS of the 1T-TaS₂ layer in the bilayer 1T/1H-TaS₂ structure for a range of d_{int} computed by using (a) pure PBE (from the manuscript), (b) PBE+U ($U_{eff}=2.27$ eV), (c) r²SCAN+rVV10, and (d) local-mBJ functionals. Contributions corresponding to majority and minority spins are illustrated as solid and dashed lines, respectively. Corrections for self-interaction error and a drawback resulting from the adoption of *averaged* exchange-correlation energy manifest in some distinct splitting strengths. However, the interplay between the 1H layer as a spin reservoir and the 1T layer as a localized spin carrier does not change.

envelope calculation to estimate the effect of screening induced by the 1-H layer. First of all, the distance between the cluster centers is about $d \approx 1.22 \text{ \AA}$, which gives an unscreened V on the order of 1 eV. The fact that there is some spectral weight on the non-central Ta atoms modifies this number very little (multipole interactions of the 6th order are small). One may think about metallic screening by the metallic 1H layer, but, as discussed, it would be incorrect. Indeed, since the charges in question are outside the 1H layer, one has to use image charges, and, as a result, contrary to what is implicitly implied in the Kondo model, the screening is not exponential but of a dipole-dipole type. Furthermore, the length of the dipole is of the same order as d , so the reduction of V due to the 1H screening is rather small.

However, one needs to take into account the intralayer 1T screening. We have calculated it in DFT+U for an undoped 1T layer. In this case the screening is semiconducting and $\epsilon_0 \approx 12$. This reduces V to less than 100 meV. However, in our case there is additional screening from mobile electrons in the lower Hubbard band, which is harder to estimate, but which is liable to add additional substantial reduction of V . Last but not least, if one does include V on a mean field level, the result is a trivial shift of the chemical potential, and the residual effect, which could, in principle, modify our calculations, only comes from residual deviations from the mean field behavior, and are expected to be rather small.

For these reasons, we believe that our neglecting of the intersite Coulomb effect is justified, especially given that our main conclusions are of qualitative nature (quantitative details anyway depend on the quality of the interface).

6. *The authors performed calculations of free-standing 1H/1T. In the experiment of Refs. 10,11,12 a substrate is present, meaning that it is assumed that the substrate does not modify the charge transfer. However, in my understanding, substrate effects can substantially change band alignment, thus modifying charge*

Figure 5: **The effect of graphene substrate on the flat bands in 1T/1H-TaS₂.** (a) Schematic illustration of stacked layers of bilayer 1T/1H-TaS₂ on multilayer graphene used in the experiment [7] (b) DOS profiles of the 1T-TaS₂ layer in 1T/1H/Gra and 1H/1T/Gra systems with a different number of graphene layers. Zero (0) graphene layers indicate the free-standing bilayer 1T/1H-TaS₂. Optimized interlayer distances between 1T and 1H layers in the two systems are 5.78 Å and 5.81 Å, respectively, which have not changed considerably from the free-standing 1T/1H bilayer with 5.81 Å of separation. When the bilayer TaS₂ is placed on a graphene substrate that provides electrons to the bilayer, the degenerate empty flat bands shift slightly toward the Fermi level, but the electron doping provided by the multilayer graphene substrate is not sufficient to compensate the strong charge transfer from the 1T to the 1H layer.

transfer. Could the authors show some theoretical evidence that a substrate would not change charge transfer in this case?

We thank the reviewer for this question. As the reviewer mentions, in practice, supporting platforms such as multilayer graphene [7, 8] or bulk 2H-TaS₂ crystal [9] are experimentally used. First, it is evident that bulk 2H-TaS₂ beneath monolayer

1T-TaS₂ has the same work function as monolayer 1H-TaS₂ as a potential electron reservoir. We investigated therefore the effect of a graphene substrate on the flat bands in 1T/1H-TaS₂. As a 5×5 supercell of graphene commensurates with a $\sqrt{13} \times \sqrt{13}$ bilayer 1T/1H-TaS₂ with 2% of lattice constant mismatch, we used the lattice constant of free-standing bilayer 1T/1H-TaS₂ when constructing the 1T-1H-multilayer graphene (or 1H-1T-graphene) stacked layer systems. Fig. 5 depicts the DOS profiles of the 1T-TaS₂ layer in the complex structure. As the graphene layers serve as an electron dopant for the bilayer, placing the bilayer on a graphene substrate shifts the degenerate empty flat bands towards the Fermi level. Although a thicker multilayer graphene substrate provides more electrons to a bilayer system, the degenerate empty flat bands do not shift further, and we did not observe any peak splitting coming from electron doping. However, it should be noted that the peak shift differs significantly between two stacking orders. When the 1T layer is positioned between the 1H layer as an electron reservoir and the graphene substrate as an electron dopant, a larger peak shift is observed than when the 1T layer is placed above the 1H layer and far from the graphene layers. This result indicates that the flat band splitting in 1T-TaS₂ can be controlled not only by the interlayer distance, but also by considering adequate substrate platforms if sufficient electrons are supplied to cause the splitting (i.e. to dope the flat band). We have added a discussion on the role of the substrate in the revised version of the manuscript and supplementary information.

7. *In Ref. 10 a gap is observed when tunneling onto the 1H layer. Is this gap also rationalized in terms of the calculations of the authors, or would the picture of doped Mott only account for the zero energy peak when the 1T layer is probed?*

In reference 10 the authors comment that the observed gap could in principle be

of a superconducting origin. In our calculations we restrict ourselves to the normal state.

To summarize, I think that their results are of interest to the first principles 2D materials community and can be of relevance for experiments in the field. Their manuscript provides a potentially useful reference for future first principles theoretical treatments, and therefore I believe that it will be a valuable addition to the literature. Given all the points above, if the authors address the points mentioned above, I would recommend their manuscript for acceptance in Nature Communications.

We thank the reviewer for their assessment and for the comments that we now addressed in the revised version of the manuscript.

REPLY TO REVIEWER 2

The authors investigate the properties of heterogeneous Ta-dichalcogenide bilayer addressing the character of the metallic state obtained interfacing 1T-TaCh₂ insulating and 1H-TaCh₂ metallic layer. Recent experiments claimed that Kondo screening leads to the screening of the local moments of the strongly correlated insulating layer due to the delocalized electrons giving rise to an heavy Fermi liquid. This observation is contrasted by recent studies of in 3D heterostructure consisting of alternating layers of 1T-TaS₂ and 1H-TaS₂ that shows strong charge transfer from the insulating layer to the metallic one. This effect induces a considerable number of holes in the 1T-TaS₂ layer reducing the role of correlation.

The paper provides a detailed analysis based on combined DFT+DMFT simulations to settle these contrasting results. The paper confirms the importance of the interlayer charge transfer, finding that the role of the metallic layer is the one of screening the local moments but rather to provide an electronic reservoir leading to a strongly doped Mott insulator.

The paper is well written, the periodic Anderson model proposed to model the bilayer is interesting and the results sound physically correct.

We thank the reviewer for their positive assessment.

However, I do have several comments that I wish the authors to consider. Please see the detailed comments below.

- 1. The strong valence fluctuations leading to a Mott insulator with less than 1/2 electron per site suggests that the interlayer Coulomb interaction between delocalized carriers in the 1H and localized one in the 1T layer plays a crucial role. Did you consider what is the role of interlayer Coulomb interaction? Do you*

have any estimate of the value of the interlayer interaction?

We thank the reviewer for the question. What induces a Mott insulator with a filling of less than $1/2$ electron per site is the charge transfer between the 1T and the 1H layers. The interlayer Coulomb interaction, as the reviewer mentions, is important, as it affects the charge transfer, and is fully taken into account in our DFT calculations.

- 2. Related to point (1). The nearest-neighbor Coulomb interaction leads to the formation of interlayer exciton complexes [electron in 1H bound to hole in the 1T layer] that as the interlayer distance is reduced might lead to the charge transfer instability. Can you comment on the role of these interlayer excitations?*

Excitons are formed between a maximum in an occupied band (where a hole is created) and a minimum in an unoccupied band (where an electron is excited). An insulating or semiconducting band structure is necessary for this mechanism. 1H TaS₂ is a good metal, so that excitons are not expected to arise, though it might be possible to specifically engineer samples that feature inter-layer excitons.

- 3. Concerning Fig.3 DMFT simulations are performed at $CT=0$ [filling 1 of the flat band in 1T layer] and $CT=1/2$. Generically, the number of electrons in the 1T layer is not conserved due to the interlayer hybridization. How did you fix the relative filling of the two layers?*

We consider the 1T electrons to be correlated and the 1H electrons to be delocalized. Hence, in the framework of DFT+DMFT, we can tune a double-counting term, effectively an orbitally-selective chemical potential, to account for an energy offset between the correlated and uncorrelated subspaces. In practice, the overall and correlated-orbital-specific chemical potential terms act as two Lagrange multipliers to fix the total and relative occupations of the layers. The double-counting

shift can operationally be obtained in various ways according to assumptions on the system at hand (fully-localized, around-mean-field, etc.). We choose to tune the double-counting potential as follows: at each DMFT step, the partial density of the correlated orbital (1T layer) is obtained by integrating the spectral function up to a maximum value, which is set to the double-counting potential. This can then be numerically tuned to reflect the desired occupation, which in turn relates to the value of the charge-transfer. The overall chemical potential, related to the total occupation, is similarly tuned in this self-consistent way.

4. *Concerning your discussion on artificial Kondo physics realized by stacking different layer of TMDs there have been recent theoretical proposals and experimental realizations of heavy Fermi liquid like behavior in TMD moire' semiconductors, see Ref. below:*

(a) <https://journals.aps.org/prb/abstract/10.1103/PhysRevB.106.L041116>

(b) <https://journals.aps.org/prresearch/abstract/10.1103/PhysRevResearch.3.043173>

(c) <https://www.science.org/doi/abs/10.1126/sciadv.ade7701>

(d) <https://www.nature.com/articles/s41586-023-05800-7>

To broader the discussion and make connection with recent advances in the field I suggest the authors to cite the aforementioned references.

We thank the reviewer for the citations which we have now added in the revised version of the manuscript.

REPLY TO REVIEWER 3

The present work provides insights into the physics of 1T/1H TaCh₂ heterostructures, exploring the possibility of classifying them as either heavy fermions or doped Mott insulators. The authors highlight two crucial competing parameters, namely the charge transfer and interlayer hybridization strength, and demonstrate through DFT calculations that these parameters vary with changes in interlayer distance. Furthermore, they propose that this system should be regarded as a highly doped Mott insulator rather than a heavy fermion, offering a clearer understanding of the experimental observations on 1T/1H TaCh₂ heterostructures. However, further investigations are required to determine the optimal parameters that accurately describe the Kondo behavior of this system. While this work contributes to enhancing our understanding of correlation effects induced by vdW heterostructures, it is suggested that publication in a more specialized journal would be appropriate. Several comments are provided to elucidate the authors' work:

We thank the reviewer for the comments.

- 1. It would be beneficial to elucidate the underlying cause of the change in charge transfer with varying interlayer distance. As my understanding, charge transfer tends to approach from 0.3 at infinite distance to 1.0 as the distance decreases, at the same time interlayer interaction strengthens. This suggests the possibility of a bonding/antibonding relationship between the two layers. Is it feasible to identify the primary characteristic of the interlayer interaction?*

This issue was briefly addressed in the paper already, as we wrote:

This behavior can be understood from the fact that there are two factors contributing to charge transfer: First, there is a chemical potential mismatch that favors some charge flow from the 1T to 1H layer; this factor is rather weakly d_{int} -dependent. At

$d_{int} \gtrsim 6.5 \text{ \AA}$ it is essentially the only factor. However, at small distances, there is substantial hybridization between the flat band of the 1T-layer and the conduction band of the 1H-layer (whose center of gravity, as seen from Fig. 2 **a, b**, is below the flat band), which pushes the flat band up (Fig. 2 **c**). This effect is, on the contrary, strongly d_{int} -dependent and kicks in for $d_{int} \lesssim 6.5 \text{ \AA}$.

We emphasize that at really infinite distances the two layers are completely decoupled and there cannot be any charge transfer. The reason is, as usual, the planar-plate capacitor's energy that grows with distance as σd_{int} , where σ is the surface charge density induced by the charge transfer. At very large distances this energy prevents the difference in the work function transfer the charge between the two layers. In the revised version, we augmented the paragraph above to be clearer, to read:

*This behavior can be understood from the fact that at moderate distances there are two factors contributing to charge transfer: First, there is a chemical potential (work function) mismatch that favors some charge flow from the 1T to 1H layer; this factor is rather weakly d_{int} -dependent. At $d_{int} \gtrsim 6.5 \text{ \AA}$ it is essentially the only factor. However, at small distances, there is substantial hybridization between the flat band of the 1T-layer and the conduction band of the 1H-layer (whose center of gravity, as seen from Fig. 2 **a, b**, is below the flat band), which pushes the flat band up (Fig. 2 **c**). This effect is, on the contrary, strongly d_{int} -dependent and kicks in for $d_{int} \lesssim 6.5 \text{ \AA}$. Finally, for $d_{int} \gg 6.5 \text{ \AA}$ (several times greater), the simple planar-plate capacitor effect, which adds additional energy cost of σd_{int} , where σ is the surface charge density induced by the charge transfer, arrests /prevents any charge transfer, despite the difference in the chemical potentials, and the bilayer, for all intents and purposes, becomes two uncoupled single layers.*

in Fig. 6 we illustrate that at $d_{int} \rightarrow \infty$ the charge transfer slowly reduced to zero and in Fig. 7 we show the results of the electrostatic potential calculations at various interlayer distances which confirm a *rather weakly d_{int} -dependent* work function in

this system. We have included these results in the supplemental information of the revised manuscript.

- 2. The physical range of charge transfer and interlayer hybridization is obtained solely through DFT calculations. However, these values can be easily influenced by different calculation options. For instance, it would be valuable to know if the authors considered full structural relaxation even at short interlayer distances, as the formation of SoD may be weakened or disrupted in highly doped cases during DFT calculations. Additionally, long-range correlation effects are known to impact band positions and charge transfer.*

We thank the reviewer for the question. We have performed very careful atomic relaxations on all structures at all distances. For our DFT and DMFT calculations, we employed the optimized structures with entirely relaxed interatomic bondings and lattice constants, but fixed the interlayer distances. The only constraint was the positioning of Ta atoms on each 1T and 1H layer in two parallel planes. For example, in the case of $d_{int}=5.8 \text{ \AA}$, Ta atoms on 1T and 1H layers are fixed at $z=0.0 \text{ \AA}$ and $z=5.8 \text{ \AA}$, respectively. However, the x and y components of Ta atoms were freely relaxed to permit CDW modulation, and the x , y , and z components of all S atoms were also freely optimized. The result is shown in Fig. 8(c). In addition, the atomic displacement in CDW modulation when the interlayer separation is varied, was investigated. Because the 3×3 charge order in the 1H-TaS₂ has a negligible impact on the charge transfer between layers (please see the answer to reviewer 1), we have focused on the formation of SoD on the 1T layer using a $\sqrt{13} \times \sqrt{13}$ supercell. Nevertheless, we have optimized 1H layers to obtain relaxed structures under the given conditions. For greater interlayer distances $d_{int} > 7.0 \text{ \AA}$, the lattice constant was fixed at 11.964 \AA of converged value. As depicted in Fig. 8(d), Ta atoms on the 1T layer form SoD even at a short interlayer distance of $d_{int}=5.5 \text{ \AA}$ by being slightly

Figure 6: Interlayer distance-dependent DOS and charge transfer in the 1T/1H-TaS₂ bilayer up to $d_{int}=10.0$ Å.

Figure 7: **1T- and 1H-side work function of bilayer 1T/1H-TaS₂ varied by interlayer distance.** Work function, a difference between Fermi level and vacuum electrostatic potential, was calculated in the framework of DFT. Due to the fact that two layers are placed in one unit cell, in the DFT calculation, they share a Fermi level but show different vacuum electrostatic potential at both surfaces. The 1T-side of the bilayer shows a steeper decrease of work function than the 1H-side when $d_{int} < 6.1 \text{ \AA}$. The work function of the 1T-side increases as the flat-band filling factor increases, whereas the 1H-side exhibits converged work function. At $d_{int} > 7.5 \text{ \AA}$, the two layers show converged values.

Figure 8: **Formation of the Star-of-David on the 1T layer in bilayer 1T/1H-TaS₂ with varying interlayer separation.** (a) The unit cell of $\sqrt{13} \times \sqrt{13}$ bilayer 1T/1H-TaS₂ structure is shown by black dashed lines. Ta atoms on 1T and 1H layers are illustrated in blue and gray, respectively. (b) Atomic displacements of Ta on 1T layer in CDW modulation, whose direction is represented by black arrows. (c) Energy-lattice constant relation in $\sqrt{13} \times \sqrt{13}$ bilayer 1T/1H-TaS₂ at fixed interlayer separations. Each energy curves are vertically offset for clarity and shown in arbitrary units. Energy minima is depicted as large filled circles, and the global energy minima at 5.8 Å of interlayer distance, of our all explored geometries, is marked with a black edge. (d) Atomic displacements of Ta on the 1T and 1H layers as a function of interlayer distance. The average deviations from a perfect triangular lattice are shown with 95% level of confidence. As interlayer spacing widens and the hybridization decreases, the 1H layer reverts to a triangular lattice, whereas the 1T layer shrinks significantly more.

shifted from the triangular lattice. The degenerate empty localized states, caused by this minor shift at $d_{int}=5.5 \text{ \AA}$ are observed above the Fermi level (*comment: original PBE+D3 result). As the interlayer distance increases, the hybridization decreases and the size of the SoD reduces as the displacement increases. The SoD size has converged at $d_{int} \approx 7.5 \text{ \AA}$. We have added the details of the calculations in the Supplemental information of the manuscript.

3. *Based on Figure 3, it seems that the susceptibilities for ($CT=1/2$, $V1=30$) and ($CT=0$, $V1=100$) exhibit similar trends upon rough comparison. Moreover, the spectral functions in Figure 4 demonstrate similar behavior, albeit with some asymmetry. If we consider different values of CT and $V1$ that exhibit similar low-energy behavior, is it possible to determine the specific values of CT and $V1$ through experimental observations? It is crucial to clarify whether this system can be classified as a heavy fermion, highly doped Mott insulator, or something in between.*

We thank the reviewer for this comment, which entails an important discussion. While it is true that the behavior of the susceptibilities for $CT = 1/2$, $V1 = 30meV$ and $CT = 0$, $V1 = 100meV$ seems to be qualitatively similar, a big difference between the two scenarios can be evidenced from the comparison of the instantaneous local moment, shown in the inset of Fig. 3 for $\tau = 0$: in fact, in the presence of charge transfer ($CT = 1/2$) its value is already dramatically screened (less than 0.5, inset Fig. 3 a), while it is considerably less so in the $CT = 0$ (no charge transfer) case (inset Fig. 3 b). X-ray absorption spectroscopy could be a way to probe the local moment experimentally.

A major difference between the doped Mott ($CT = 1/2$, $V1 = 30meV$) and heavy Fermion scenario ($CT = 0$, $V1 = 100meV$) appears in the *momentum* dependent spectral functions ($A(\mathbf{k}, \omega)$), which are experimentally measurable via angular

Figure 9: Comparison of total, 1T and 1H Fermi surface and momentum-resolved spectral function $A(\mathbf{k}, \omega)$, for the two parameters sets $CT = 1/2$, $V1 = 30$ meV and $CT = 0$, $V1 = 100$ meV at temperatures $T = 116$ K and 9.66 K.

resolved photoemission spectroscopy (ARPES). We show total and layer resolved spectral functions at temperatures $T = 116$ K and 9 K in Fig. 9. A clear distinction between the heavy Fermion scenario can be seen from the spectral functions along a high-symmetry path and most clearly from the spectral weight maps at the Fermi energy ($A(\mathbf{k}, \omega = 0)$).

In the ($CT = 0$, $V1 = 100meV$)-case we find at the higher temperature incoherent spectral weight at the Fermi level originating from the T-layer and from the H-layer bands at the Fermi level. When coherence in the T-layer develops at low temperature the apparent Fermi surface completely reconstructs with new Fermi surface segments originating from coherent quasiparticles appearing in the T-layer. The strong hybridization between T and H-layer states can be inferred from avoided crossings of the low-energy quasi particle bands and from the character change (T-layer vs H-layer) along the Fermi surface segments appearing at low temperature. Such a temperature induced reconstruction of the Fermi segments is indeed characteristic of heavy Fermion systems.

The ($CT = 1/2$, $V1 = 30meV$) case differs from this scenario. While some sharpening of spectral features upon lowering the temperature appears also here, we observe at the Fermi level T-layer-derived coherent spectral weight already at the higher temperature and there is no complete reconstruction as is the case for $CT = 0$, $V1 = 100meV$. This can be most clearly seen in the H-layer Fermi level spectral weight, which remains almost unchanged with temperature ($CT = 1/2$, $V1 = 30meV$)-case — in strong contrast to the ($CT = 0$, $V1 = 100meV$)-case.

This is why we think that ARPES (possibly nano-ARPES given the typical system size) might be a good way to distinguish doped Mott and heavy Fermion scenarios. We added these spectral functions to the revised supplemental information and added a corresponding discussion also to the revised main text.

-
- [1] J. A. Wilson, F. J. Di Salvo, and S. Mahajan, Charge-density waves in metallic, layered, transition-metal dichalcogenides, *Phys. Rev. Lett.* **32**, 882 (1974).
- [2] J. A. Wilson, More concerning cdw phasing in 2h-tase₂, *Journal of Physics F: Metal Physics* **15**, 591 (1985).
- [3] Z. Wang, Y.-Y. Sun, I. Abdelwahab, L. Cao, W. Yu, H. Ju, J. Zhu, W. Fu, L. Chu, H. Xu, and K. P. Loh, Surface-limited superconducting phase transition on 1t-tas₂, *ACS Nano* **12**, 12619 (2018).
- [4] J. Ning, M. Kothakonda, J. W. Furness, A. D. Kaplan, S. Ehlert, J. G. Brandenburg, J. P. Perdew, and J. Sun, Workhorse minimally empirical dispersion-corrected density functional with tests for weakly bound systems: r²SCAN + rVV10, *Phys. Rev. B* **106**, 075422 (2022).
- [5] T. Rauch, M. A. L. Marques, and S. Botti, Local modified becke-johnson exchange-correlation potential for interfaces, surfaces, and two-dimensional materials, *Journal of Chemical Theory and Computation* **16**, 2654 (2020), pMID: 32097004, <https://doi.org/10.1021/acs.jctc.9b01147>.
- [6] P. Darancet, A. J. Millis, and C. A. Marianetti, Three-dimensional metallic and two-dimensional insulating behavior in octahedral tantalum dichalcogenides, *Physical Review B* **90**, 045134 (2014).
- [7] V. Vaño, M. Amini, S. C. Ganguli, G. Chen, J. L. Lado, S. Kezilebieke, and P. Liljeroth, Artificial heavy fermions in a van der waals heterostructure, *Nature* **599**, 582 (2021).
- [8] W. Wan, R. Harsh, A. Meninno, P. Dreher, S. Sajan, I. Errea, F. de Juan, and M. M. Ugeda, *Magnetic order in a coherent two-dimensional Kondo lattice* (2022), arXiv:2207.00096 [cond-mat].
- [9] C. G. Ayani, M. Pisarra, I. M. Ibarburu, M. Garnica, R. Miranda, F. Calleja, F. Martín,

and A. L. V. de Parga, Two-dimensional Kondo lattice in a TaS₂ van der Waals heterostructure (2022), arXiv:2205.11383 [cond-mat].

REVIEWER COMMENTS

Reviewer #1 (Remarks to the Author):

I thank the authors for the detailed response to my comments. I note below a few follow-up comments motivated by their response, using the same numbers the authors used in their response:

1) The results of the authors are slightly surprising, given that there is substantial experimental evidence suggesting that Se-S replacements are the drivers of the charge transfer in 4Hb TaS₂. This could be, of course, a limitation of the DFT formalism in this material. To address this point, I would suggest that the authors add a statement in the manuscript such as “While chalcogen replacements are likely a driving force for a change in the filling in 4Hb TaS₂, our DFT calculations do not find substantial changes in the charge transfer. This could hint at a potential limitation of bare DFT calculation to account for charge transfer in these materials.”.

3) I thank the authors for noting their argument, yet I would like to respectfully point out that I do not fully agree with their argument regarding energy scales. Hubbard U in Ta can be on the order of 1.5-2 eV, which implies that each CDW orbital is affected by that U through an approximate participation ratio of 13, yielding an effective U in the Wannier orbital of 100 meV. This would imply a relatively sizable change in the band alignment. To address this point, I would suggest that the authors add the following statement to their manuscript: “Our calculations do not include possible DFT+U corrections, which can potentially change the band alignment and charge transfer in the system.”

7) Ref. 10 shows that the gap cannot be of superconducting origin, as it persists up to large out-of-plane magnetic fields. Given that the gap of Ref. 10 is not superconducting, could the authors comment if their picture would support the existence of such a gap? From my understanding, the doped Mott regime proposed would be inconsistent with the presence of a gap in the 1H layer, and in the doped Mott regime, the 1H would be metallic. To address this point, I would ask the authors to include the following statement in their revised manuscript: “We note that the doped Mott regime found in our calculations is incompatible with the gap observed in 1H in Ref. 10, suggesting that those bilayers on HOPG cannot be rationalized within the doped Mott scenario.”

Finally, in order to incorporate these aspects in the abstract, I would suggest that the authors make the following minor replacements in their abstract:

- “based on first-principles calculations which indicates” → “based on first-principles calculations in free-standing bilayers which indicates”

- “We accurately quantify the strength of the interlayer hybridization which allows us to unambiguously determine that the system is much closer to a doped Mott insulator than to a heavy fermion scenario” → “We quantify the strength of the interlayer hybridization, which allows us to suggest that free-standing bilayers would be close to a doped Mott insulator regime.”

To summarize, I think that the authors put forward an interesting first-principles study of dichalcogenide bilayers. I will recommend acceptance in Nature Communications once the authors implement the changes outlined above.

Reviewer #2 (Remarks to the Author):

The authors addressed all my comments, but still there are some arguments presented in the work that are not fully convincing and deserve further clarification.

1) Coming back to question #1 of my previous report.

The reply is not completely clear, since it does not clarify how the long-range interaction is taken into account in the DMFT calculations presented in the manuscript.

The nearest-neighbor interlayer interaction is particularly important to determine the energetics of the charge transfer between the two layer and the estimate of its value should be provided in the manuscript. Moreover, the interaction could also favor interlayer electron-hole excitonic complexes changing the qualitative results presented in the manuscript.

2) It is not clear why the Mott insulator persist down to $CT=1/2$?

Decreasing the filling in the 1T layer should reduce correlations and lead to metallic behavior. Thus, for this large value of the charge transfer one would rather expect a metallic behavior in both layers.

Reviewer #3 (Remarks to the Author):

The following points need to be clarified for the referee report:

1. Regarding the charge transfer variation, the authors mentioned that the charge transfer at short distances is induced by the upshift of flat bands. However, it's intriguing that such a sensitive upshift occurs due to weak van der Waals interlayer interactions. Could the authors specify which orbitals in 1H-TaS₂ are interacting with the flat band of 1T-TaS₂? Based on Figure 6 in the authors' report, it appears that the flat band is the most sensitive to shifting, while other bands remain relatively insensitive.

2. The formation of stable SoD at short interlayer distances ($\sim 5.5 \text{ \AA}$) is noteworthy. It's particularly interesting that SoD can remain stable with one less electron, as traditional CDW systems are typically sensitive to electron doping. Is there a specific reason why SoD remains stable under electron doping in this case?

3. The authors suggest additional experiments to distinguish between the heavy fermion and doped Mott scenario. It is agreed that ARPES or local moments could be valuable experimental tools to confirm the scenario. However, this also implies that the scenario cannot be definitively confirmed based solely on the limited DFT (Density Functional Theory) and DMFT (Dynamical Mean-Field Theory) calculations. Thus, it remains an open question, and further work in this area is warranted.

In summary, if the issues raised in points 1 and 2 can be clearly explained, this work could be recommended for publication in this journal. However, the need for additional experimental validation, as mentioned in point 3, highlights the ongoing nature of this research.

LIST OF CHANGES

1. Added Supplementary Note 5 containing a discussion on the possible effects of non-local interaction terms. Added a sentence in the main text to refer to it (lines 244-246).
2. Modified the discussion of the DFT+U contribution in the Methods section of the main text (lines 430-437) to include a discussion of the work function difference.
3. Expanded a sentence in the main text to properly convey that the spectral features alone are not sufficient to discriminate between different types of physics at play (lines 358-362).
4. Added Supplementary Note 7 detailing the different d-orbitals contribution to inter-layer hybridization. Added a sentence in the main text to refer to it (line 452).

REPLY TO REVIEWER 1

I thank the authors for the detailed response to my comments. I note below a few follow-up comments motivated by their response, using the same numbers the authors used in their response:

We thank the reviewer for their assessment. Below, we answer point by point their observations.

- 1. The results of the authors are slightly surprising, given that there is substantial experimental evidence suggesting that Se-S replacements are the drivers of the charge transfer in 4Hb TaS₂. This could be, of course, a limitation of the DFT formalism in this material. To address this point, I would suggest that the authors add a statement in the manuscript such as “While chalcogen replacements are likely a driving force for a change in the filling in 4Hb TaS₂, our DFT calculations do not find substantial changes in the charge transfer. This could hint at a potential limitation of bare DFT calculation to account for charge transfer in these materials.”*

With regards to the role of Se replacement in the physics of charge transfer, we should point out that the reviewer’s statement — “given that there is substantial experimental evidence suggesting that Se-S replacements are the drivers of the charge transfer in 4Hb TaS₂” — does not align with the recent experimental reports. As far as we are aware, there is a recent report [1] from Weizmann Institute. Binghai Yan is one of the co-authors in both our manuscript and this experimental report. In the experimental paper, the authors speculated that Se-S replacement may stop the charge transfer while clean 4Hb TaS₂ always exhibits charge transfer. This is opposite to what the Reviewer claimed. Our work deals with the clean 4Hb TaS₂ and the DFT-calculated charge transfer is consistent with the Weizmann Institute

experiments. Understanding whether Se matters, which is still inconclusive in the experimental paper, is beyond the scope of our present work, although it might be an interesting question.

- 3. I thank the authors for noting their argument, yet I would like to respectfully point out that I do not fully agree with their argument regarding energy scales. Hubbard U in Ta can be on the order of 1.5-2 eV, which implies that each CDW orbital is affected by that U through an approximate participation ratio of 13, yielding an effective U in the Wannier orbital of 100 meV. This would imply a relatively sizable change in the band alignment. To address this point, I would suggest that the authors add the following statement to their manuscript: “Our calculations do not include possible DFT+ U corrections, which can potentially change the band alignment and charge transfer in the system.”*

We thank the reviewer for the observation, but we respectfully disagree with their assessment: first, it is important to note that DFT+ U is mostly reliable in the cases of atomic orbitals, and it is known to present issues when dealing with an electron distributed over several lattice sites, as already established for example in the paradigmatic case of VO_2 [2] and organic materials [3].

Nevertheless, we did perform LDA+ U calculations (as detailed in Supplementary Note 5) , and, as expected, found no meaningful change in the charge transfer. Since we include the Hubbard U in our model calculations, we, of course, used in the latter the input from straight DFT.

We agree with the reviewer on the order of magnitude of the effective Hubbard U , which we also value at $\approx 100\text{meV}$. On the matter of band stacking, this value has to be compared with the bonding/anti-bonding gap of the charge density wave of the T electrons, in absolute value $\approx 600\text{meV}$ as visible from Fig. 1b in the main text.

We can also assess the effects of this term on charge transfer. We can define a comparative energy scale through the work function differences between the 1T and 1H layers. Our calculations (cfr. Supplementary Table 1) show that regardless of CDW formation the 1H layer has a work function that is approximately 800 meV larger than that of the 1T layer. This explains the propensity of electrons to be transferred from the T layer to the H. We agree with the reviewer that DFT+U type corrections related to $U \approx 100$ meV could affect also the work functions by ≈ 100 meV. However, the bonding/anti-bonding energy gap is 6 times larger than this contribution, and the work function difference is 8 times larger. This is why we do not expect a U correction as implemented in DFT+U to provide significant alterations to our picture.

To sum up, DFT+U corrections are not expected to strongly affect the band stacking and charge transfer. We have rephrased the comment on DFT+U in the Methods section of the main text in order to better address the reviewer’s observations.

7. *Ref. 10 shows that the gap cannot be of superconducting origin, as it persists up to large out-of-plane magnetic fields. Given that the gap of Ref. 10 is not superconducting, could the authors comment if their picture would support the existence of such a gap? From my understanding, the doped Mott regime proposed would be inconsistent with the presence of a gap in the 1H layer, and in the doped Mott regime, the 1H would be metallic. To address this point, I would ask the authors to include the following statement in their revised manuscript: “We note that the doped Mott regime found in our calculations is incompatible with the gap observed in 1H in Ref. 10, suggesting that those bilayers on HOPG cannot be rationalized within the doped Mott scenario.”*

We agree with the reviewer that the doped Mott regime would be at odds with the presence of a gap in the 1H layer. We however argue that the interpretation of

the experimental data of ref. 10 is rather delicate. While some of the features shown by the samples considered in ref. 10 are indeed compatible with the Kondo insulator scenario proposed by the authors (such as the presence of a gap in the 1H electron DOS), others seem to be less easily reconcilable: for example, the experimental data of fig. 2c show no hint of the presence of an indirect hybridization gap down to 300 mK in the 1T layer. Indeed, there are different possible causes for a spectral weight depletion occurring in the 1H layer at the Fermi level. One possible explanation close to the Kondo insulator / heavy fermion picture of ref. 10, but also in line with the scenario of a doped Mott insulator, would be the Fano effect.

The Fano effect has been invoked to explain asymmetric as well as symmetric resonances and also spectral weight depletions / “dips” at the Fermi level measured in STM experiments with adatoms on surfaces [4, 5]. While these can be of Kondo origin, the Fano resonance mechanism itself only necessitates the hybridization of the conduction electrons with a generic narrow quasiparticle peak at the Fermi level. This scenario is fully compatible with our doped Mott theoretical characterization.

However, other explanations of a gap at the Fermi level of the 1H layer are also possible. These include CDW effects and still also claims of unconventional superconductivity [6]. As a consequence, we relied on other features (e.g. the local moment behavior) to support our theoretical picture.

Since the emergence of a spectral weight depletion does not allow to distinguish between Kondo insulator and doped Mott scenarios, and furthermore the exact origin of the depletion in the 1H layer appears as a rather open issue, we think that adding the strong statement suggested by the reviewer to our paper would not be correct. In the manuscript, we stated *the evolution of a narrow coherence peak observed in spectra cannot alone distinguish between the doped Mott and the heavy fermion scenario*. We have now expanded the previously quoted sentence to more clearly address the difficulty of drawing conclusions from spectral features alone.

Finally, in order to incorporate these aspects in the abstract, I would suggest that the authors make the following minor replacements in their abstract:

We detail the answers to each point

8. *“based on first-principles calculations which indicates” → “based on first-principles calculations in free-standing bilayers which indicates”*

We point out to the reviewer that we also considered non-free-standing samples, as detailed in Supplementary Note 2, where the effect of different substrates has been discussed and deemed not determinant to the overall physics. Therefore, we respectfully disagree with the proposed rephrasing.

9. *“We accurately quantify the strength of the interlayer hybridization which allows us to unambiguously determine that the system is much closer to a doped Mott insulator than to a heavy fermion scenario” → “We quantify the strength of the interlayer hybridization, which allows us to suggest that free-standing bilayers would be close to a doped Mott insulator regime.”*

As discussed in the previous point, our analysis is not limited to the free-standing case. We don't think the rephrasing, or the removal of the adverb “accurately” is therefore warranted.

To summarize, I think that the authors put forward an interesting first-principles study of dichalcogenide bilayers. I will recommend acceptance in Nature Communications once the authors implement the changes outlined above.

We thank the reviewer for their overall positive assessment, and we hope to have exhaustively answered to their observations.

REPLY TO REVIEWER 2

The authors addressed all my comments, but still there are some arguments presented in the work that are not fully convincing and deserve further clarification.

We thank the reviewer and address the points raised below

- 1. Coming back to question #1 of my previous report. The reply is not completely clear, since it does not clarify how the long-range interaction is taken into account in the DMFT calculations presented in the manuscript. The nearest-neighbor interlayer interaction is particularly important to determine the energetics of the charge transfer between the two layer and the estimate of its value should be provided in the manuscript. Moreover, the interaction could also favor interlayer electron-hole excitonic complexes changing the qualitative results presented in the manuscript.*

We thank the reviewer for their observation. We are convinced that the in-plane long range interaction, which was not taken into account, can quantitatively, but not qualitatively change the results. Mean-field interlayer Coulomb is fully accounted for in the underlying DFT. Effects beyond DFT, such as interlayer excitons, are only observed in large-gap semiconductors, like $\text{WSe}_2/\text{MoSe}_2$, and even there they are hard to detect. We have added Supplementary Note 5 detailing our considerations on long-range interaction terms.

- 2. It is not clear why the Mott insulator persist down to $CT=1/2$? Decreasing the filling in the 1T layer should reduce correlations and lead to metallic behavior. Thus, for this large value of the charge transfer one would rather expect a metallic behavior in both layers.*

We thank the reviewer for the question, which gives us the opportunity to clarify what is perhaps a slightly misleading -though standard- terminology: while a Mott insulator is in fact insulating, a *doped* Mott insulator is metallic. In our case, the system is expected to be a strongly-correlated metal. Indeed, as we assess in the main text, *the charge transfer between 1T-TaS₂ and 1H-TaS₂ drives the Mott insulating state in 1T-TaS₂ far away from the half-filling regime (one electron per correlated orbital) of the 1T single layer, introducing itinerant charge carriers inside the 1T-TaS₂ layer.*

REPLY TO REVIEWER 3

The following points need to be clarified for the reviewer report:

We answer each point below

1. *Regarding the charge transfer variation, the authors mentioned that the charge transfer at short distances is induced by the upshift of flat bands. However, it's intriguing that such a sensitive upshift occurs due to weak van der Waals interlayer interactions. Could the authors specify which orbitals in 1H-TaS₂ are interacting with the flat band of 1T-TaS₂? Based on Figure 6 in the authors' report, it appears that the flat band is the most sensitive to shifting, while other bands remain relatively insensitive.*

We thank the reviewer for the pertinent question. The binding has a considerable vdW contribution, but it has still enough covalent character to bind the 3D TaS₂ even without the vdW correction (as opposed to, say, graphite). Among the 1H-TaS₂ orbitals, it is the out-of-plane d_{z^2} orbital that provides the larger overlap, as detailed in the Methods section. To confirm this, we can introduce an estimator for the hybridization of different orbitals defined as

$$\Omega(\phi_\alpha, \phi_\beta) := |\langle \phi_\alpha | \Psi_{nk} \rangle| \cdot |\langle \phi_\beta | \Psi_{nk} \rangle|$$

where $\phi_{\alpha,\beta}$ are the chosen d orbitals and Ψ_{nk} is the Kohn-Sham wavefunction near the Fermi level. The results are shown in Fig. 1, where the orbital characters corresponding to the largest values of Ω are plotted. As expected, the overlap between the d_{z^2} orbitals in the 1H and 1T layers is noticeably larger than the others, including the second largest ($d_{x^2-y^2}(H), d_{z^2}(T)$). We have included the picture and a brief explanation in Supplementary Note 7.

Figure 1. **Hybridization estimator between different orbitals.** Orbital hybridization estimator for an inter-layer distance of (a) 5.8 Å and (b) 7.0 Å. The panels on the left replicate Fig. 2a,b in the manuscript at the two chosen interlayer distances. On the right we show the DFT band structure weighted with the hybridization estimator Ω for the largest contributions to the overall hybridization.

2. *The formation of stable SoD at short interlayer distances (5.5 Å) is noteworthy. It's particularly interesting that SoD can remain stable with one less electron, as traditional CDW systems are typically sensitive to electron doping. Is there a specific reason why SoD remains stable under electron doping in this*

case?

We thank the reviewer for the interesting point, which is suitable to be expanded upon in future work. We can give a first qualitative explanation: the model consists of 6 bonding orbitals, 6 anti-bonding and 1 non-bonding (the flat band). Since the 6 bonding orbitals are well separated in energy from the non-bonding, even when the latter is depleted the SoD pattern remains robust, as it can be evinced from Fig. 1b in the main text.

- 3. The authors suggest additional experiments to distinguish between the heavy fermion and doped Mott scenario. It is agreed that ARPES or local moments could be valuable experimental tools to confirm the scenario. However, this also implies that the scenario cannot be definitively confirmed based solely on the limited DFT (Density Functional Theory) and DMFT (Dynamical Mean-Field Theory) calculations. Thus, it remains an open question, and further work in this area is warranted.*

We agree with the reviewer that this area of research warrants further analysis. In any event, we believe the theoretical evidence is strong, and in fact our study is only theoretical. The techniques we employed, such as DMFT, offer a clear argument to discriminate between the possible scenarios, given their well-proven ability to assess the presence or absence of well-formed local moments at various temperatures. The experimental scenarios we proposed are in our opinion adequate tools to confirm our hypothesis.

In summary, if the issues raised in points 1 and 2 can be clearly explained, this work could be recommended for publication in this journal. However, the need for additional experimental validation, as mentioned in point 3, highlights the ongoing nature of this research.

We thank the reviewer for their positive assessment and we hope to have addressed their concerns.

-
- [1] A. K. Nayak, A. Steinbok, Y. Roet, J. Koo, I. Feldman, A. Almoalem, A. Kanigel, B. Yan, A. Rosch, N. Avraham, and H. Beidenkopf, First order quantum phase transition in the hybrid metal-mott insulator transition metal dichalcogenide 4hb-TaS₂ (2023), arXiv:2303.01447 [cond-mat.str-el].
 - [2] D. Wickramaratne, N. Bernstein, and I. I. Mazin, Role of defects in the metal-insulator transition in VO₂ and V₂O₃, Phys. Rev. B **99**, 214103 (2019).
 - [3] J. Ferber, K. Foyevtsova, H. O. Jeschke, and R. Valentí, Unveiling the microscopic nature of correlated organic conductors: The case of κ -(et)₂Cu[n(cn)₂]br_xcl_{1-x}, Phys. Rev. B **89**, 205106 (2014).
 - [4] V. Madhavan, W. Chen, T. Jamneala, M. F. Crommie, and N. S. Wingreen, Tunneling into a single magnetic atom: Spectroscopic evidence of the kondo resonance, Science **280**, 567 (1998).
 - [5] J. Li, W.-D. Schneider, R. Berndt, and B. Delley, Kondo scattering observed at a single magnetic impurity, Phys. Rev. Lett. **80**, 2893 (1998).
 - [6] V. Vaño, S. C. Ganguli, M. Amini, L. Yan, M. Khosravian, G. Chen, S. Kezilebieke, J. L. Lado, and P. Liljeroth, Evidence of nodal superconductivity in monolayer 1h-TaS₂ with hidden order fluctuations, Advanced Materials **n/a**, 2305409.

REVIEWERS' COMMENTS

Reviewer #1 (Remarks to the Author):

The authors have addressed the different comments in my report. I would like to thank them for their detailed response. There are some minor points that could be discussed related to the reliability of DFT in making their statements, and from my perspective, it would be informative for the readers that those limitations are stated explicitly. Nonetheless, for the sake of moving forward, I believe that their current version is at the stage that can be accepted in Nature Communications despite some methodological overstatements.

I would finally comment on their explanation for the appearance of a gap in the 1H side of the heterostructure. The authors mention that it could stem from Fano resonance. However, this would require a local $S=1/2$, which corresponds to the opposite scenario of the one proposed in the manuscript. The authors also mention that such a gap in 1H can be related to CDW or unconventional superconductivity. As stated in my previous report, the fact that the gap survives large out-of-plane magnetic fields shows that it is not superconducting. CDW orders in TMDC materials are not capable of fully gapping the Fermi surface, and no additional CDW was observed experimentally. Given those points, I would judge that their explanation for the existence of the gap in 1H is not compatible with the doped Mott scenario proposed by the authors. Of course, it is perfectly fine that their theoretical analysis is not compatible with experiments, as their results are clearly of interest to theorists in the field. However, such inconsistency should be acknowledged in their manuscript or the SI to avoid misleading readers.

To summarize, once the authors address that final point, the manuscript can be accepted in Nature Communications.

Reviewer #2 (Remarks to the Author):

The authors have addressed all my comments. I think the current version of the manuscript is suitable for publication in nature communications.

Reviewer #3 (Remarks to the Author):

Most of my questions are well responded.

Although there are still several issues in the origin of the charge transfer depending on the interlayer distance and the stable SoD in highly doped case, it should be studied in further studies.

So, let me recommend this manuscript for publication as it is.

LIST OF CHANGES

1. Rephrased a passage in the main text (line 360-369) to evidence the presence in ref. 10 of a spectral weight dip in the 1H layer, the explanation of which is not conclusively provided in this paper.

REPLY TO REVIEWER 1

The authors have addressed the different comments in my report. I would like to thank them for their detailed response. There are some minor points that could be discussed related to the reliability of DFT in making their statements, and from my perspective, it would be informative for the readers that those limitations are stated explicitly. Nonetheless, for the sake of moving forward, I believe that their current version is at the stage that can be accepted in Nature Communications despite some methodological overstatements.

We thank the reviewer for their positive assessment of our response and our work in general. Though we are confident of the suitability of the used methods, we appreciate the reviewer's concerns. In this context, the topic will certainly benefit from further investigations, both theoretical and experimental.

I would finally comment on their explanation for the appearance of a gap in the 1H side of the heterostructure. The authors mention that it could stem from Fano resonance. However, this would require a local $S=1/2$, which corresponds to the opposite scenario of the one proposed in the manuscript. The authors also mention that such a gap in 1H can be related to CDW or unconventional superconductivity. As stated in my previous report, the fact that the gap survives large out-of-plane magnetic fields shows that it is not superconducting. CDW orders in TMDC materials are not

capable of fully gapping the Fermi surface, and no additional CDW was observed experimentally. Given those points, I would judge that their explanation for the existence of the gap in 1H is not compatible with the doped Mott scenario proposed by the authors. Of course, it is perfectly fine that their theoretical analysis is not compatible with experiments, as their results are clearly of interest to theorists in the field. However, such inconsistency should be acknowledged in their manuscript or the SI to avoid misleading readers.

We thank the reviewer for their comment. We believe the Fano resonance mechanism can be used to discuss a very broad range of “dip + peak” phenomena, not necessarily requiring localized spin momenta. However, we appreciate the experimental evidence provided in Ref. 10 is at odds with a CDW or superconducting scenario, and compatible with a Kondo lattice description as well. We have rephrased a passage in the main text to reflect the interpretation of the observations from a spectral perspective.

To summarize, once the authors address that final point, the manuscript can be accepted in Nature Communications.

We thank the reviewer.

REPLY TO REVIEWERS 2 AND 3

We thank the reviewers for their positive assessment.